# Kinematics-Driven Gaussian Shape Deformation for Blurry Monocular Dynamic Scenes

Yeon-Ji Song [1 2]   Kiyoung Kwon [1]   Junoh Lee [3]   Jin-Hwa Kim [1 4 †]   Byoung-Tak Zhang [1 2 †]

## Abstract

Reconstructing dynamic 3D scenes from blurry monocular videos is challenging as motion-induced blur entangles object motion and geometry, hindering geometric consistency. We present Kinematics-GS, a kinematics-aware framework that models blur as motion-aligned deformation and introduces a kinematic prior to reparameterize Gaussian shapes along motion trajectories, thereby mitigating degenerate shape collapse without auxiliary motion supervision. To stabilize optimization, we decompose scenes into dynamic and static components using temporal deformation variance and employ a coarse-to-fine deformation strategy to capture both global motion and fine-grained details. We also introduce a challenging real-world dataset of deformable and elastic objects exhibiting non-rigid motion with spatially non-uniform motion blur that obscures geometric cues. Extensive experiments on real-world benchmarks with realistic motion blur demonstrate that Kinematics-GS outperforms prior methods by a clear margin in monocular dynamic scene reconstruction, highlighting its effectiveness in handling complex and non-rigid motion scenarios.

## 1. Introduction

Real-world scenes are intrinsically dynamic, comprising static structures and continuously moving objects. Reconstructing such environments from visual observations remains a fundamental challenge, as accurately capturing object geometry and motion from monocular input is inherently ambiguous. This ambiguity complicates the separation of static backgrounds and dynamic foregrounds, particularly in the presence of non-rigid motion. The problem is further exacerbated by *motion blur*, where object motion is temporally integrated during camera exposure, smearing high-frequency details along motion trajectories and obscuring precise spatial cues. As a result, robust dynamic scene reconstruction requires not only reliable static–dynamic separation but also effective mitigation of motion-induced degradation under unconstrained, real-world conditions.

Recent advances in 3D Gaussian Splatting (3DGS) (Kerbl et al., 2023) have emerged as a compelling alternative to NeRF (Mildenhall et al., 2021), enabling explicit scene representations with high-fidelity rendering. Building on these capabilities, 3DGS has been extended to dynamic settings for time-varying geometry and scene reconstruction (Lu et al., 2025; Bui et al., 2025; Zhu et al., 2024; Wu et al., 2024; Yang et al., 2024b). Complementary structural priors, such as anchor-based representations (Li et al., 2025; Zhu et al., 2025; Lei et al., 2025) and Level-of-Detail (LoD) hierarchies (Liu et al., 2026; Kulhanek et al., 2025; Ren et al., 2024), further improved representation efficiency and optimization stability, allowing scalable dynamic reconstruction without sacrificing visual quality.

Despite these advancements, existing dynamic 3DGS methods face several limitations that hinder their robustness in real-world settings. First, many approaches are predominantly evaluated on synthetic or simplified benchmarks, which fail to capture the complexity and variability of real-world motion, thereby limiting generalization. Second, prior methods typically assume sharp input images and accurate camera poses (Fu et al., 2024), frequently relying on auxiliary networks such as pre-trained optical flow estimators (Lu et al., 2025). This assumption breaks down in real-world handheld scenarios, where motion blur degrades pose estimation and leads to floating artifacts and temporal inconsistencies. Third, monocular dynamic reconstruction is inherently ill-posed, and optimization based solely on visual cues such as flow consistency (Yang et al., 2024a; Bae et al., 2024) is often unstable. Without explicit physical regularization, deformation models are prone to ambiguous or implausible 3D motion. As a result, existing methods struggle to simultaneously disentangle static and dynamic

---

[†]Corresponding authors.

[1]AI Institute, Seoul National University, [2]Interdisciplinary Program in Neuroscience, Seoul National University, [3]Gwangju Institute of Science and Technology, [4]NAVER AI Lab. Correspondence to: Jin-Hwa Kim <j1nhwa.kim@navercorp.com>, Byoung-Tak Zhang <btzhang@snu.ac.kr>.

*Proceedings of the $43^{rd}$ International Conference on Machine Learning*, Seoul, South Korea. PMLR 306, 2026. Copyright 2026 by the author(s).

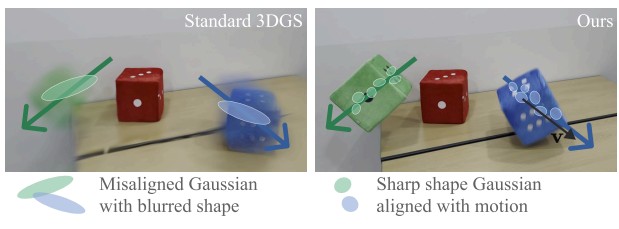

*Figure 1.* Comparison between standard 3D Gaussian Splatting and our method under motion blur. Our approach aligns Gaussian primitives with the estimated velocity direction **v**, producing sharp and physically consistent shapes. Arrows indicate motion direction, while ellipses visualize Gaussian anisotropy.

components and maintain temporally coherent reconstruction under realistic motion blur, leaving monocular reconstruction in real-world dynamic scenes an open challenge.

To address these challenges, we present **Kinematics-GS**, a kinematics-aware framework for dynamic scene reconstruction from blurry monocular videos. The proposed approach explicitly disentangles static and dynamic Gaussians, focusing modeling capacity on regions with genuine temporal variation while preserving static scene structure. To handle the multi-scale nature of dynamic motion, a kinematics-guided covariance regularization is introduced, which constructs a local motion-aligned basis from instantaneous velocity and reparameterizes Gaussian anisotropy to enforce deformation consistent with the underlying kinematics, as illustrated in Fig. 1. This physically motivated prior resolves monocular motion ambiguities and mitigates geometric oversimplification, thereby enabling stable optimization and high-fidelity reconstruction under realistic motion blur.

Our main contributions are as follows:

- We introduce a 3D Gaussian representation that decouples static and dynamic components using temporal deformation variance, without auxiliary segmentation.

- We propose a kinematics-guided covariance refinement that couples object motion with Gaussian shape to resolve monocular ambiguities under motion blur.

- We present a real-world dataset composed of deformable and elastic objects, designed to benchmark robustness in monocular dynamic scene reconstruction under non-rigid motion and severe motion blur.

## 2. Related Work

### 2.1. Dynamic Scene Reconstruction

Recovering dynamic 3D scenes from monocular video is an inherently ill-posed problem due to the ambiguity between geometry, camera motion, and object deformation.

While multi-view (Bansal et al., 2020; Li et al., 2022) and stereo (Attal et al., 2020) camera setups mitigate these ambiguities via explicit geometric constraints, monocular settings require strong regularization priors. Early NeRF-based approaches incorporated auxiliary cues such as predicted depth (Du et al., 2021; Park & Kim, 2024), optical flow (Li et al., 2021; Zhou et al., 2024), or data-driven priors, including object-centric motion representations (Luthra et al., 2024; Pumarola et al., 2021; Song et al., 2023). However, these methods often suffer from slow optimization speeds and limited generalizability, as they rely heavily on the quality of off-the-shelf estimators, which degrade rapidly under motion blur or occlusion (Li et al., 2024). Although recent works integrate camera pose estimation (Fu et al., 2024; Bui et al., 2025) or deblurring modules (Chen & Liu, 2024; Lee et al., 2024a), they typically address either camera motion or object motion in isolation. Consequently, achieving robust reconstruction in unconstrained handheld videos, where camera shake and fast object motion coexist, remains a significant challenge.

### 2.2. Dynamic Gaussian Splatting

Leveraging the efficiency of 3DGS, recent works have extended Gaussian splatting to dynamic scenes, which can be broadly categorized into 4D primitives and deformation-based approaches. 4D methods lift Gaussians into spacetime using Hex-plane encoders (Cao & Johnson, 2023; Duan et al., 2024; Wu et al., 2024) or 4D spheroids (Yang et al., 2024b), achieving high rendering quality but often struggling with large, non-linear motions. Deformation-based methods (Yang et al., 2024a; Bae et al., 2024) instead model motion by displacing canonical Gaussians via learnable fields. However, their under-constrained nature frequently leads to ambiguities between static and dynamic components, resulting in floaters or distorted geometry (Wang et al., 2025; Wu et al., 2025; Lee et al., 2024b). Recent works address blurry monocular inputs through sparse controls or blur-aware modeling (Song et al., 2025b), demonstrating promising results on real-world sequences, yet are limited to physically grounded motion constraints. To improve appearance stability, prior methods introduce regularization based on optical flow (Zhu et al., 2024) or local rigidity (Luthra et al., 2024), while kinematic or physics-based priors have been explored mostly in object-centric (Song et al., 2025a) or category-specific settings (Zhou et al., 2025). In our work, we introduce a kinematics-guided regularization tailored to general deformation-based 3DGS, enforcing stability and consistency without restricting representational flexibility.

### 2.3. Level of Detail

Level of Detail (LoD) techniques optimize rendering performance by adapting representation complexity based on viewing distance (Luebke et al., 2002). In the context of

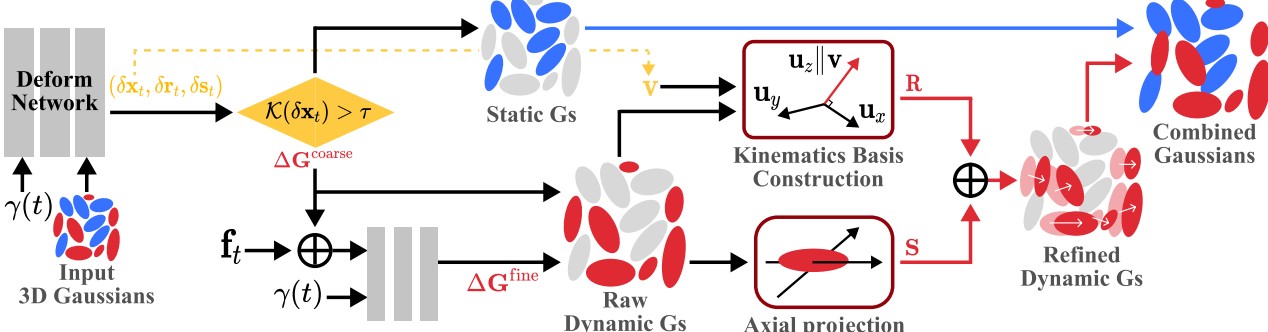

*Figure 2.* **Overview of our pipeline.** Given canonical 3D Gaussian primitives initialized from an SfM reconstruction of a blurry monocular input video, a deformation network predicts per-Gaussian offsets $(\delta\mathbf{x}_t, \delta\mathbf{r}_t, \delta\mathbf{s}_t)$ conditioned on time $t$. Based on the temporal variance $\mathcal{K}(\delta\mathbf{x}_t)$ of predicted positional offsets, the scene is decomposed into static and dynamic Gaussian sets. For dynamic primitives, a coarse-to-fine deformation strategy is employed, where a neighborhood-aggregated coarse deformation offsets $\Delta\mathbf{G}^{\text{coarse}}$ captures stable low-frequency motion, followed by fine-grained residual refinement $\Delta\mathbf{G}^{\text{fine}}$ to recover distinctive dynamics. Subsequently, a kinematics-guided covariance refinement module constructs a local motion-aligned basis $(\mathbf{u}_x, \mathbf{u}_y, \mathbf{u}_z)$ from the velocity $\mathbf{v}$, and refines the rotation and scale factors by restricting deformation to physically plausible directions. The refined dynamic Gaussians are unified with static Gaussians to form the combined Gaussian representation.

3DGS, LoD strategies have been employed to prune redundant Gaussians or adjust spatial hierarchies for real-time inference (Seo et al., 2025; Yang et al., 2025; Kulhanek et al., 2025). However, these methods are predominantly designed for static scenes. Directly applying LoD structures to dynamic scenes is non-trivial, as temporal inconsistencies often lead to severe flickering artifacts. To address this, our work integrates a continuous coarse-to-fine deformation strategy into a flexible LoD framework. By explicitly modeling spatially coherent motion patterns before refining them with fine-grained residuals, we effectively stabilize the optimization landscape, preventing the local minima issues common in existing frameworks.

## 3. Preliminaries

### 3.1. 3D Gaussian Splatting

3DGS (Kerbl et al., 2023) represents a scene as a collection of 3D Gaussians. Each Gaussian is parameterized by a position $\mathbf{x} \in \mathbb{R}^3$, opacity $\alpha$, view-dependent spherical harmonics (SH) coefficients $c$, and a 3D covariance matrix $\Sigma$. The covariance matrix is factorized into a rotation matrix $R \in \mathbb{R}^{3\times3}$ and a diagonal scaling matrix $S = \text{diag}(\mathbf{s}) \in \mathbb{R}^{3\times3}$, such that $\Sigma = RSS^\top R^\top$. For differentiable rasterization, each 3D Gaussian is projected onto the image plane, with resulting 2D covariance matrix $\Sigma'$:

$$\Sigma' = JW\Sigma W^\top J^\top, \tag{1}$$

where $W$ denotes the linear component of the viewing transformation matrix and $J$ is the Jacobian of the affine perspective projection. Then, the final pixel colors are obtained by sorting Gaussians along the viewing direction and composit-

ing them via volumetric $\alpha$-blending:

$$\widehat{C} = \sum_{i=1}^{N} c_i \alpha_i' \prod_{j=1}^{i-1}(1 - \alpha_j'), \tag{2}$$

where $\alpha_i'$ denotes the projected opacity of the $i$-th Gaussian and $c_i$ is the view-dependent color from its SH coefficients.

### 3.2. Level-of-Detail 3DGS

We adopt the importance pruning mechanism from Flexible Level of Detail (FLoD) (Seo et al., 2025), which extends 3DGS with a hierarchical multi-resolution representation. FLoD enforces a level-dependent lower bound on the Gaussian scale to control representation granularity. At level $l \in \{1, \ldots, L_{\max}\}$, the minimum scale is defined as:

$$s_{\min}^{(l)} = \begin{cases} \lambda\rho^{1-l}, & l < L_{\max} \\ 0, & l = L_{\max} \end{cases} \tag{3}$$

where $\lambda$ denotes the initial scale constraint and $\rho \in (0,1)$ controls its geometric decay across levels. The Gaussian scale is parameterized as $s^{(l)} = \exp(s_{\text{opt}}) + s_{\min}^{(l)}$, enforcing a level-dependent minimum while allowing unconstrained refinement at the finest level.

Training proceeds in a coarse-to-fine manner. At each level, Gaussians with low accumulated importance are pruned, while optimized Gaussians are cloned to initialize the next level with scale reparameterization for continuity.

## 4. Method

We propose a robust monocular dynamic scene reconstruction framework that integrates kinematic principles into the

3DGS pipeline. The core challenge lies in motion blur, which embeds object motion directly into spatial observations and renders deformation-based reconstruction severely under-constrained. Our approach explicitly addresses this ill-posed nature by decomposing the scene into static and dynamic components and applying a kinematics-guided regularization that aligns Gaussian primitives with physical motion trajectories. An overview is illustrated in Fig. 2.

### 4.1. Dynamic-Static Decomposition

Dynamic scenes contain a mixture of rigid background and deformable objects. Since kinematic modeling is only meaningful for entities that undergo physical motion, we first decouple dynamic primitives from the static background.

Given a canonical Gaussian $\mathcal{G}$ with position $\mathbf{x}$, rotation $\mathbf{r}$, and scaling $\mathbf{s}$, a deformation network $\mathcal{F}_\theta$ predicts per-Gaussian offsets at time $t$:

$$(\delta\mathbf{x}_t, \delta\mathbf{r}_t, \delta\mathbf{s}_t) = \mathcal{F}_\theta(\mathbf{x}, \mathbf{r}, \mathbf{s}, \gamma(t) + \epsilon), \qquad (4)$$

where $\gamma(t)$ denotes temporal positional encoding, and $\epsilon \sim \mathcal{N}(0, \sigma_\epsilon^2)$ is the linearly decaying Gaussian noise annealed during training (Yang et al., 2024a) (see Appendix A.2).

Dynamic primitives are identified by analyzing the temporal variance of position offsets over a window of $T$ frames. Specifically, deformation variance score $\mathcal{K}$ is formulated as:

$$\mathcal{K}(\{\delta\mathbf{x}_t\}_{t=1}^T) = \frac{1}{T}\sum_{t=1}^T \|\delta\mathbf{x}_t - \delta\bar{\mathbf{x}}_t\|^2, \ \delta\bar{\mathbf{x}}_t = \frac{1}{T}\sum_{t=1}^T \delta\mathbf{x}_t. \qquad (5)$$

Gaussians with $\mathcal{K}$ exceeding a threshold $\tau$, determined via the ablation results in Tab. 3, are classified as the dynamic set $\mathcal{G}_d$, while the remainder form the static set $\mathcal{G}_s$. This decomposition mitigates monocular ambiguity by preventing static regions from absorbing dynamic deformation, providing a stable initialization for subsequent motion-aware Gaussian optimization. We denote the kinematic deformation of dynamic Gaussians as $\Delta\mathbf{G} = (\delta\mathbf{x}_t, \delta\mathbf{r}_t, \delta\mathbf{s}_t)$.

### 4.2. Coarse-to-Fine Deformation

Directly optimizing high-resolution Gaussian deformation under severe blur is prone to local minima. We therefore adopt a coarse-to-fine strategy, capturing global motion at low resolution first and progressively refining fine-grained non-rigid deformations. This approach recovers local details and corrects boundary artifacts, effectively reducing ambiguities from motion blur and monocular supervision.

Initially, a coarse deformation field aggregates neighboring dynamic Gaussians to enforce spatial coherence:

$$\Delta\mathbf{G}^{\mathrm{coarse}} = \frac{1}{|\mathcal{N}|}\sum_{j\in\mathcal{N}} \Delta\mathbf{G}_j, \qquad (6)$$

where $\mathcal{N} \subset \mathcal{G}_d$ denotes the $k$-nearest dynamic neighbors of $\mathbf{G}$. This approach stabilizes optimization by capturing dominant low-frequency motion patterns.

The coarse deformation enforces a stable motion coherence but inevitably smooths out locally distinctive dynamics. To recover fine-grained details, residual deformations are predicted conditioned on a learnable per-Gaussian dynamic feature vector $\mathbf{f}_t$ and the temporal encoding $\gamma(t)$:

$$\Delta\mathbf{G}^{\mathrm{fine}} = g_\phi(\mathbf{f}_t, \gamma(t)), \qquad (7)$$

where $g_\phi$ is a shared MLP applied to dynamic primitives. The final deformation is obtained by residual composition of the coarse and fine components:

$$\Delta\mathbf{G} = \Delta\mathbf{G}^{\mathrm{coarse}} + \Delta\mathbf{G}^{\mathrm{fine}}. \qquad (8)$$

### 4.3. Kinematic Basis Construction

Motion blur is modeled in a physically meaningful manner by constructing a local kinematic coordinate frame for each dynamic Gaussian based on its predicted translational motion. This basis serves as a motion-aligned reference frame, enabling anisotropic shape deformation to directions consistent with the underlying object kinematics.

The instantaneous velocity $\mathbf{v}$ of a dynamic primitive is computed from its predicted translation displacement offset $\delta\mathbf{x}_t$ over the frame interval $\Delta t$. The primary motion axis $\mathbf{u}_z$ is then obtained by normalizing the velocity direction:

$$\mathbf{u}_z = \frac{\mathbf{v}}{|\mathbf{v}|}, \ \text{where} \ \mathbf{v} = \frac{\delta\mathbf{x}_t}{\Delta t}. \qquad (9)$$

To construct a stable orthonormal basis, the $\mathbf{u}_x$ axis is determined by computing the cross product of the primary motion axis $\mathbf{u}_z$ and a canonical reference direction $\mathbf{r}_{\mathrm{ref}}$. This operation yields a direction orthogonal to the motion:

$$\mathbf{u}_x = \frac{\mathbf{u}_z \times \mathbf{r}_{\mathrm{ref}}}{\|\mathbf{u}_z \times \mathbf{r}_{\mathrm{ref}}\|}. \qquad (10)$$

When $\mathbf{r}_{\mathrm{ref}}$ becomes nearly colinear with $\mathbf{u}_z$, an alternative canonical reference direction is used to avoid degeneracy. The remaining axis is then defined via the cross product:

$$\mathbf{u}_y = \mathbf{u}_z \times \mathbf{u}_x. \qquad (11)$$

This construction yields a deterministic motion-aligned coordinate frame that varies smoothly with the predicted velocity. The resulting rotation matrix $\widetilde{\mathbf{R}}$, or *kinematic basis*, is:

$$\widetilde{\mathbf{R}} = [\mathbf{u}_x \mid \mathbf{u}_y \mid \mathbf{u}_z] \in SO(3). \qquad (12)$$

By aligning one axis with the instantaneous velocity, this basis provides a physically grounded reference frame for modeling motion-induced anisotropic deformation. Please refer to Appendix A.1 for further details.

## 4.4. Kinematics-Guided Covariance Refinement

In dynamic 3D Gaussian representations, conventional co-variance parameterizations become unstable under motion blur, as unconstrained anisotropic deformation can lead to spurious elongation or geometric collapse. To address this limitation, the covariance of dynamic primitives is refined using a motion-aligned kinematic prior that constrains deformation along velocity, enforcing kinematically consistent anisotropy while preserving geometry orthogonal to motion.

Given the motion-aligned basis $\widetilde{\mathbf{R}}$ derived in Sec. 4.3, the predicted covariance $\Sigma$ is first evaluated along each basis onto this local frame to extract motion-aware variances:

$$\sigma_k^2 = \mathbf{u}_k^\top \boldsymbol{\Sigma}\, \mathbf{u}_k, \quad \text{for } k \in \{x, y, z\}. \tag{13}$$

Here, $\sigma_x$ and $\sigma_y$ represent the intrinsic object geometry, remaining invariant to temporal motion integration, while $\sigma_z$ captures the spatial extent along the motion direction. $\boldsymbol{\Sigma}$ denotes the raw covariance predicted by the deform network, which is constrained to be symmetric positive-definite.

These components are reparameterized into axis-aligned Gaussian scales to explicitly model motion blur as:

$$s_x' = \sigma_x, \quad s_y' = \sigma_y, \quad s_z' = \sigma_z + \eta\, \|\mathbf{v}\|\, \Delta t, \tag{14}$$

where the longitudinal scale $s_z'$ integrates physical displacement over the exposure time, and $\|\mathbf{v}\|\Delta t$ represents the spatial displacement during exposure.

An alignment factor $\eta \in [0, 1]$ modulates this blur magnitude based on the consistency between the predicted deformation orientation $\mathbf{r}_z$ and the velocity direction $\mathbf{u}_z$:

$$\eta = \max\left(|\, \mathbf{r}_z \cdot \mathbf{u}_z\,|,\ \sigma(\kappa)\right), \tag{15}$$

where $\mathbf{r}_z$ denotes the principal axis of the predicted deformation rotation, and $\sigma(\kappa)$ is a sigmoid-based lower bound that ensures numerical stability under noisy velocity estimates. This formulation effectively suppresses spurious elongation caused by noisy estimates, while preserving physically valid motion blur aligned with the velocity direction.

To allow for fine-grained shape adjustment beyond the kinematic prior, a learnable residual $\delta\mathbf{s}$ is applied in log-space:

$$\boldsymbol{\ell}_s = \log\left(\mathbf{s}'\right) + \lambda_s \delta\mathbf{s}, \tag{16}$$

where $\mathbf{s}' = [\, s_x', s_y', s_z'\,]^\top \in \mathbb{R}^3$ and $\lambda_s = 0.1$. Here, $\log(\cdot)$ denotes the element-wise logarithm applied to the scale vector, and $\lambda_s$ controls the residual magnitude. The final scaling matrix is then obtained as:

$$\mathbf{S} = \mathrm{diag}(\exp(\boldsymbol{\ell}_s)). \tag{17}$$

In parallel, the rotational component is updated by composing the motion-aligned kinematic basis with a learnable residual rotation $\delta\mathbf{r}$:

$$\mathbf{R} = \widetilde{\mathbf{R}} \exp(\delta\mathbf{r}), \tag{18}$$

where $\exp(\delta\mathbf{r}) \in SO(3)$ denotes the rotation matrix obtained via the exponential map from the residual rotation vector $\delta\mathbf{r} \in \mathbb{R}^3$. The residual rotation accounts for local deviations not captured by the velocity-aligned basis. The final kinematics-guided covariance is reconstructed as:

$$\boldsymbol{\Sigma}_{\mathrm{kin}} = \mathbf{R}\,\mathbf{S}\,\mathbf{S}^\top \mathbf{R}^\top. \tag{19}$$

This formulation guarantees a valid symmetric positive-definite covariance while explicitly encoding motion-aligned anisotropy through the kinematic rotation and scale parameterization. As a result, dynamic Gaussians elongate consistently with the underlying motion trajectory, mitigating spurious deformation under motion blur. The refined dynamic Gaussians are then combined with the static ones and jointly rendered, enabling consistent appearances while concentrating motion modeling on dynamic regions.

## 4.5. Model Training

We train our framework end-to-end using a composite objective function designed to balance high-fidelity photometric reconstruction with geometric regularization. The overall training objective is defined as follows:

$$\mathcal{L} = \mathcal{L}_{\mathrm{img}} + \lambda_{\mathrm{reg}}\mathcal{L}_{\mathrm{reg}} + \lambda_{\mathrm{ani}}\mathcal{L}_{\mathrm{ani}}. \tag{20}$$

Following standard 3DGS practice, the photometric loss combines $\mathcal{L}_1$ and D-SSIM:

$$\mathcal{L}_{\mathrm{img}} = (1 - \lambda_{\mathrm{dssim}})\|I - I_{gt}\|_1 + \lambda_{\mathrm{dssim}}(1 - \mathrm{SSIM}(I, I_{gt})). \tag{21}$$

To ensure physically plausible dynamics, we impose constraints on the predicted position offsets $\delta\mathbf{x}$. Static and dynamic regularizations penalize the magnitude of position offsets for $\mathcal{G}_s$ and $\mathcal{G}_d$, encouraging static consistency and controlled motion:

$$\mathcal{L}_{\mathrm{reg}} = \frac{1}{|\mathcal{G}_s|} \sum_{i \in \mathcal{G}_s} \|\delta\mathbf{x}_i\|_2 + \frac{1}{|\mathcal{G}_d|} \sum_{j \in \mathcal{G}_d} \|\delta\mathbf{x}_j\|_2, \tag{22}$$

where $|\mathcal{G}_s|$ and $|\mathcal{G}_d|$ denote the cardinality of the static and dynamic Gaussian sets, respectively.

An anisotropy loss $\mathcal{L}_{\mathrm{ani}}$ is introduced to prevent degenerate Gaussians with extreme aspect ratios during deformation:

$$\mathcal{L}_{\mathrm{ani}} = \frac{1}{N} \sum_{i=1}^{N} \frac{\max(\mathbf{s}_i)}{\min(\mathbf{s}_i) + \epsilon_{\mathrm{ani}}}, \tag{23}$$

where $\epsilon_{\mathrm{ani}}$ is a small constant to ensure numerical stability.

## 5. Experiments

We evaluate our framework on diverse real-world dynamic scenes with motion blur. We first describe the experimental

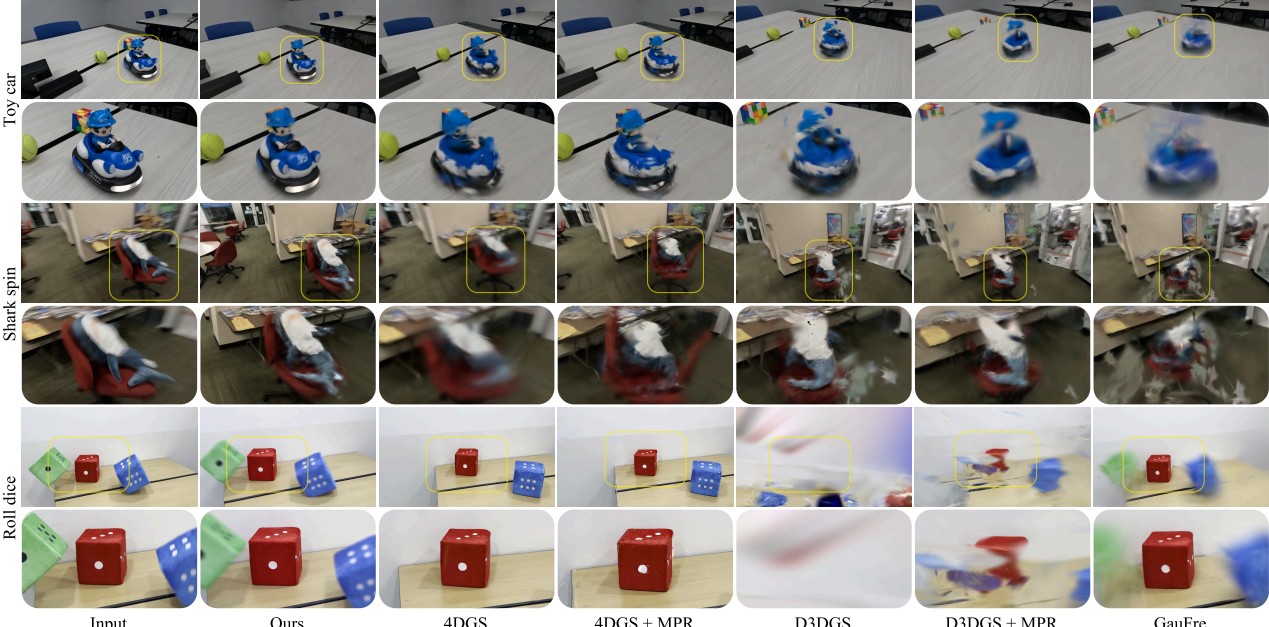

*Figure 3.* Qualitative results on BARD-GS and DEOs real-world blurry datasets. Compared to baseline methods, our approach produces cleaner reconstructions with fewer motion-induced artifacts, preserving object shape and structural integrity under severe blur. Complete qualitative comparisons are provided in the supplementary material.

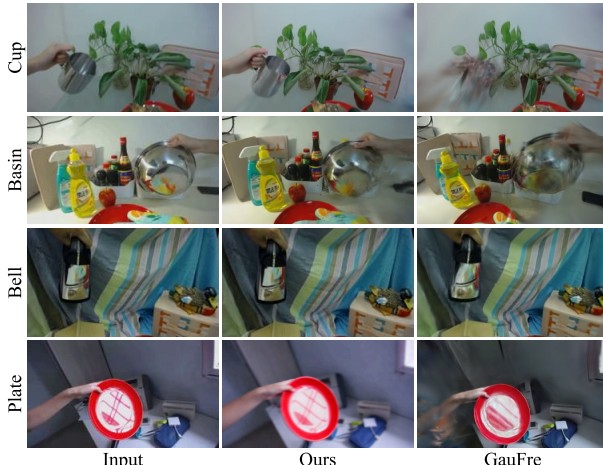

*Figure 4.* Dynamic specular object reconstruction on the NeRF-DS dataset. Although the baseline methods achieve comparable results, our method preserves both geometric structure and specular appearance under motion, capturing characteristic view-dependent reflections and reducing dynamic-region artifacts.

details and datasets, followed by quantitative and qualitative comparisons with state-of-the-art methods. Finally, we present ablation studies analyzing the impact of individual components and key hyperparameters.

### 5.1. Implementation Details

Our framework is implemented in PyTorch and trained on a single NVIDIA RTX 3090 GPU. We follow the multi-resolution training strategy of FLoD (Seo et al., 2025), progressively optimizing the scene from LoD 1 to LoD 5 over a total of 30k iterations. All quantitative results are reported using the final reconstruction at LoD 5. Unless otherwise specified, all hyperparameters are fixed across experiments. Additional training details are provided in Appendix B.

### 5.2. Datasets

**Real-world blurry dataset.** We evaluate our method on eight scenes from the BARD-GS dataset (Lu et al., 2025), which captures complex dynamic motions with significant motion blur using synchronized GoPro cameras. The dataset provides motion-blurred frames for training and corresponding sharp images for evaluation, making it well-suited for assessing robustness under real-world dynamic conditions.

**Dynamic specular dataset.** We evaluate our method on six scenes from the NeRF-DS iPhone dataset (Yan et al., 2023), which features dynamic scenes with non-rigid motion and challenging specular effects. The dataset provides monocular video sequences with fast object motions that inherently induce a slight motion blur, serving as a standard benchmark for evaluating dynamic scene reconstruction under both appearance and motion ambiguities.

**Our custom dataset.** Existing dynamic scene benchmarks primarily focus on rigid body motions and sharp imagery captured with fast shutter speeds, failing to capture com-

*Table 1.* Quantitative comparison on two real-world blurry datasets, BARD-GS (Lu et al., 2025) and *DEOs*. We compare our method against 4DGS (Wu et al., 2024), D3DGS (Yang et al., 2024a) (with and without MPRNet (Zamir et al., 2022)), Deblur3DGS (Lee et al., 2024a), and GauFre (Liang et al., 2025). Each color indicates the best , second best , and third best respectively.

| | BARD-GS | | | | | | | | DEOs | | | |
|---|---|---|---|---|---|---|---|---|---|---|---|---|
| SCENE | TOYCAR | RUBIKCUBE | CARD | CUBEDESK | SHARKSPIN | WALK | KITCHEN | MICROLAB | ROLLDICE | REDDICE | BOUNCE | STRIKE |
| | | | | | | PSNR ↑ | | | | | | |
| 4DGS+MPR | 19.513 | 23.061 | 21.308 | 18.770 | 20.007 | 14.730 | 18.264 | 21.405 | 23.957 | 23.405 | 25.256 | 22.346 |
| 4DGS | 19.706 | 22.032 | 17.792 | 19.172 | 20.256 | 15.839 | 18.245 | 20.954 | 25.507 | 21.089 | 23.246 | 28.284 |
| D3DGS+MPR | 19.605 | 24.664 | 21.252 | 21.472 | 17.923 | 14.954 | 19.872 | 20.022 | 20.350 | 23.239 | 27.986 | 24.851 |
| D3DGS | 19.710 | 20.948 | 18.242 | 21.218 | 17.149 | 16.460 | 19.353 | 19.439 | 20.094 | 26.404 | 30.138 | 29.866 |
| DEBLUR3DGS | 21.596 | 26.160 | 16.028 | 23.326 | 16.483 | 13.495 | 18.024 | 17.902 | 22.395 | 26.040 | 26.390 | 24.023 |
| GAUFRE | 20.789 | 20.370 | 18.815 | 20.008 | 17.872 | 16.599 | 18.444 | 20.741 | 34.055 | 26.484 | 28.374 | 32.646 |
| **OURS** | 24.644 | 28.266 | 24.455 | 23.600 | 25.394 | 20.283 | 23.545 | 25.374 | 35.049 | 27.903 | 30.405 | 33.757 |
| | | | | | | SSIM ↑ | | | | | | |
| 4DGS+MPR | 0.881 | 0.891 | 0.851 | 0.774 | 0.798 | 0.668 | 0.802 | 0.786 | 0.908 | 0.836 | 0.956 | 0.920 |
| 4DGS | 0.884 | 0.878 | 0.799 | 0.773 | 0.796 | 0.690 | 0.806 | 0.773 | 0.910 | 0.821 | 0.953 | 0.955 |
| D3DGS+MPR | 0.869 | 0.902 | 0.856 | 0.814 | 0.775 | 0.701 | 0.823 | 0.746 | 0.901 | 0.834 | 0.955 | 0.938 |
| D3DGS | 0.880 | 0.865 | 0.813 | 0.807 | 0.759 | 0.712 | 0.816 | 0.738 | 0.893 | 0.845 | 0.977 | 0.961 |
| DEBLUR3DGS | 0.829 | 0.860 | 0.524 | 0.803 | 0.689 | 0.602 | 0.759 | 0.685 | 0.899 | 0.824 | 0.950 | 0.923 |
| GAUFRE | 0.885 | 0.845 | 0.809 | 0.791 | 0.751 | 0.717 | 0.774 | 0.756 | 0.966 | 0.864 | 0.974 | 0.973 |
| **OURS** | 0.936 | 0.938 | 0.902 | 0.857 | 0.898 | 0.807 | 0.887 | 0.872 | 0.989 | 0.930 | 0.977 | 0.980 |
| | | | | | | LPIPS ↓ | | | | | | |
| 4DGS+MPR | 0.150 | 0.148 | 0.185 | 0.198 | 0.210 | 0.372 | 0.286 | 0.161 | 0.150 | 0.167 | 0.051 | 0.117 |
| 4DGS | 0.148 | 0.179 | 0.327 | 0.203 | 0.205 | 0.305 | 0.276 | 0.171 | 0.061 | 0.198 | 0.069 | 0.055 |
| D3DGS+MPR | 0.197 | 0.118 | 0.216 | 0.151 | 0.355 | 0.414 | 0.249 | 0.244 | 0.155 | 0.179 | 0.086 | 0.106 |
| D3DGS | 0.153 | 0.205 | 0.340 | 0.147 | 0.347 | 0.368 | 0.259 | 0.278 | 0.157 | 0.103 | 0.033 | 0.061 |
| DEBLUR3DGS | 0.230 | 0.269 | 0.491 | 0.228 | 0.420 | 0.427 | 0.309 | 0.320 | 0.168 | 0.179 | 0.061 | 0.102 |
| GAUFRE | 0.163 | 0.238 | 0.378 | 0.231 | 0.389 | 0.370 | 0.306 | 0.230 | 0.115 | 0.230 | 0.053 | 0.085 |
| **OURS** | 0.094 | 0.118 | 0.162 | 0.151 | 0.186 | 0.322 | 0.219 | 0.123 | 0.041 | 0.094 | 0.039 | 0.048 |

*Table 2.* Quantitative comparison on NeRF-DS iPhone dataset (Yan et al., 2023), which consists of dynamic specular objects.

| | AS | | | BASIN | | | BELL | | | CUP | | | PLATE | | | PRESS | | |
|---|---|---|---|---|---|---|---|---|---|---|---|---|---|---|---|---|---|---|
| METHOD | PSNR↑ | SSIM↑ | LPIPS↓ | PSNR↑ | SSIM↑ | LPIPS↓ | PSNR↑ | SSIM↑ | LPIPS↓ | PSNR↑ | SSIM↑ | LPIPS↓ | PSNR↑ | SSIM↑ | LPIPS↓ | PSNR↑ | SSIM↑ | LPIPS↓ |
| 4DGS+MPR | 23.008 | 0.809 | 0.173 | 18.313 | 0.717 | 0.211 | 21.753 | 0.769 | 0.173 | 22.715 | 0.841 | 0.132 | 16.522 | 0.666 | 0.356 | 22.973 | 0.767 | 0.191 |
| 4DGS | 23.261 | 0.816 | 0.166 | 18.884 | 0.733 | 0.197 | 21.548 | 0.778 | 0.176 | 23.241 | 0.855 | 0.118 | 17.095 | 0.684 | 0.336 | 22.396 | 0.738 | 0.253 |
| D3DGS+MPR | 25.969 | 0.874 | 0.188 | 19.584 | 0.789 | 0.191 | 24.911 | 0.833 | 0.167 | 24.712 | 0.886 | 0.158 | 20.518 | 0.815 | 0.217 | 25.459 | 0.863 | 0.193 |
| D3DGS | 26.210 | 0.885 | 0.179 | 19.586 | 0.793 | 0.192 | 25.352 | 0.847 | 0.158 | 24.374 | 0.888 | 0.158 | 20.456 | 0.815 | 0.220 | 25.280 | 0.862 | 0.192 |
| DEBLUR3DGS | 21.049 | 0.790 | 0.183 | 17.389 | 0.702 | 0.280 | 20.208 | 0.761 | 0.189 | 21.938 | 0.840 | 0.142 | 18.039 | 0.792 | 0.318 | 22.993 | 0.798 | 0.206 |
| GAUFRE | 25.808 | 0.879 | 0.133 | 19.261 | 0.761 | 0.175 | 25.453 | 0.843 | 0.119 | 20.598 | 0.825 | 0.237 | 20.077 | 0.787 | 0.238 | 25.311 | 0.858 | 0.163 |
| **OURS** | 27.230 | 0.881 | 0.120 | 22.395 | 0.802 | 0.159 | 28.309 | 0.894 | 0.093 | 24.930 | 0.902 | 0.100 | 20.938 | 0.819 | 0.218 | 25.909 | 0.870 | 0.190 |

plex non-rigid deformations and motion blur inherent in real-world environments. To address this limitation, we introduce the **D**eformable and **E**lastic **O**bjects (**DEOs**) dataset, which comprises real-world sequences capturing complex physical behaviors such as elasticity and compression. The dataset features objects including cotton doll, foam dice, and bouncing balls, recorded across indoor and outdoor environments using an iPhone 15 Pro. Additional dataset details are provided in Appendix C. Our code and the DEOs dataset are publicly available to the research community.

### 5.3. Baselines and Metrics

We compare our method dynamic view synthesis approaches, including 4DGS (Wu et al., 2024), D3DGS (Yang et al., 2024a), Deblur3DGS (Lee et al., 2024a), and GauFre (Liang et al., 2025). Since 4DGS and D3DGS do not explicitly handle motion blur, we additionally evaluate two-stage variants of these methods, where input images are preprocessed using MPRNet (Zamir et al., 2022) for deblurring prior to reconstruction. Evaluation is conducted using standard metrics, including PSNR, SSIM, and LPIPS,

together with inference speed and the number of initialized Gaussians to assess computational efficiency.

### 5.4. Results on Reconstruction under Motion Blur

We evaluate our method on real-world blurry monocular videos, where motion blur severely degrades dynamic reconstruction. As shown in Tab. 1, our method consistently outperforms state-of-the-art approaches across all metrics, producing sharper geometry and temporally stable reconstructions under fast motion in Fig. 3. The result indicates that directly regressing covariance under motion blur leads to unstable anisotropic deformation, whereas our motion-aligned strategy yields more stable optimization by disentangling intrinsic geometry from motion-induced anisotropy.

Specifically, we compare our approach with GauFre (Liang et al., 2025), a concurrent method that also incorporates static–dynamic decomposition. While GauFre is primarily designed for real-time rendering, our method explicitly models motion blur through a kinematics-guided deformation prior. As a result, our approach more accurately isolates gen-

*Table 3.* Per-scene ablation study on two scenes from the BARD-GS and DEOs datasets. Ours trained with $\tau = 2e{-}5$. C.F. and K.R. denote the coarse-to-fine strategy and kinematics-guided regularization, respectively.

| | **TOYCAR** | | | **ROLLDICE** | | |
|---|---|---|---|---|---|---|
| **METHOD** | PSNR↑ | SSIM↑ | LPIPS↓ | PSNR↑ | SSIM↑ | LPIPS↓ |
| **OURS** | 24.644 | 0.936 | 0.094 | 32.053 | 0.948 | 0.100 |
| w/o C.F. | 24.111 | 0.929 | 0.113 | 27.790 | 0.932 | 0.144 |
| w/o K.R. | 8.152 | 0.460 | 0.543 | 17.018 | 0.882 | 0.246 |
| w/o $\mathcal{L}_{\text{REG}}$ | 24.604 | 0.935 | 0.096 | 17.466 | 0.883 | 0.246 |
| w/o $\mathcal{L}_{\text{ANI}}$ | 24.352 | 0.934 | 0.098 | 17.328 | 0.876 | 0.243 |
| $\tau = 4e{-}5$ | 23.837 | 0.924 | 0.132 | 27.902 | 0.930 | 0.124 |
| $\tau = 6e{-}5$ | 18.804 | 0.842 | 0.303 | 21.614 | 0.889 | 0.268 |
| $\tau = 8e{-}5$ | 17.353 | 0.768 | 0.429 | 20.270 | 0.869 | 0.340 |
| $\tau = 1e{-}4$ | 17.125 | 0.755 | 0.460 | 19.888 | 0.866 | 0.350 |

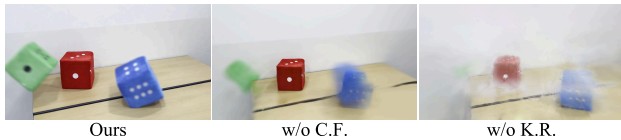

| Ours | w/o C.F. | w/o K.R. |

*Figure 5.* Roll Dice ablation example for w/o C.F. and w/o K.R.

uinely dynamic regions under blur, leading to higher-fidelity reconstructions and improved temporal stability.

### 5.5. Robustness to Appearance and Motion Complexity

We further evaluate robustness under complex appearance changes using dynamic scenes with specular objects from the NeRF-DS iPhone dataset. Table 2 demonstrates that our method achieves competitive performance across standard reconstruction metrics. However, the qualitative advantages are more pronounced in Fig. 4, where our approach exhibits substantially improved temporal visual consistency with sharper object boundaries and fewer motion artifacts compared to the closest baseline. Additional qualitative results are provided in Appendix D.

### 5.6. Computational Efficiency

Our method demonstrates high rendering efficiency, consistently achieving real-time frame rates between 200 and 700 FPS across all datasets. This corresponds to an order-of-magnitude speedup over baseline methods, reaching up to $26\times$ faster rendering on the BARD-GS dataset. Crucially, the performance is achieved with a highly compact representation, utilizing significantly fewer Gaussian primitives than the baseline models. Detailed comparisons of both runtime speed (FPS) and spatial complexity (number of Gaussians) are reported in Appendix E. Although training time increases by roughly $2\times$ due to additional computational components, this overhead yields significant gains in both visual fidelity and interactive rendering speed.

### 5.7. Ablation Studies

We conduct ablation studies to validate key design choices and analyze sensitivity to hyperparameters. Table 3 summarizes the quantitative impact of each component, while Fig. 5 provides corresponding visual comparisons. Comprehensive experimental results are available in Appendix F.

**Coarse-to-Fine strategy (C.F.).** To evaluate the effectiveness of the coarse-to-fine deformation strategy, we disable neighborhood-based coarse aggregation. Removing this component degrades reconstruction quality, indicating that motion coherence across neighboring Gaussians is essential for stable optimization and accurate deformation estimation.

**Kinematics-guided regularization (K.R.).** We assess the role of kinematics-guided regularization by removing the motion-aligned basis construction. Without this constraint, deformable Gaussians frequently degenerate into needle-like or isotropic artifacts under motion blur. Our Kinematics projection-based refinement enforces physically consistent anisotropy aligned with motion direction, leading to sharper geometry and reduced artifacts.

**Loss terms ($\mathcal{L}$).** We further analyze the contribution of individual loss terms on both BARD-GS and DEOs. Removing either the regularization loss $\mathcal{L}_{\text{reg}}$ or the anisotropy loss $\mathcal{L}_{\text{ani}}$ results in noticeable geometric degradation. These results highlight that combining photometric supervision with geometric constraints is crucial for preventing Gaussian collapse and ensuring robust optimization.

**Separation threshold ($\tau$).** Tab. 3 shows that overly small values of $\tau$ may suppress genuine dynamic regions, while overly large values may introduce unnecessary motion modeling into static backgrounds. In practice, our method exhibits stable performance across a broad range around $\tau = 2e{-}5$. This indicates low sensitivity to precise threshold selection; therefore, we fix $\tau$ in this range for all scenes.

## 6. Conclusion

In this work, we present Kinematic-GS, a novel framework that resolves the geometric ambiguities of blurry monocular video by explicitly aligning Gaussian shape deformation with kinematic trajectories. By integrating dynamic-static decomposition with the kinematic-guided geometric regularization, we successfully isolate static and dynamic Gaussians, focusing on modeling regions of temporal change. Extensive experiments demonstrate that our joint optimization strategy effectively mitigates motion ambiguities and prevents global geometric oversimplification, yielding high-fidelity scene reconstruction in unconstrained real-world scenarios compared to previous methods.

**Limitations and future work.** Despite promising results, our method has limitations that open avenues for future research. First, the kinematics-guided modules introduce a computational overhead, resulting in longer training times. Future work will focus on optimizing this via fused CUDA kernels. Second, while the LoD strategy reduces rendering cost, memory consumption for long sequences remains a challenge shared by explicit representations, which we plan to mitigate through compression techniques.

## Acknowledgements

This work was partly supported by grants from the IITP (RS-2021-II211343-GSAI/10%, RS-2022-II220951-LBA/15%, RS-2022-II220953-PICA/15%), the NRF (RS-2024-00353991-SPARC/15%, RS-2023-00274280-HEI/15%), the KEIT (RS-2025-25453780/15%), and the KIAT (RS-2025-25460896/15%), funded by the Korean government.

## Impact Statement

This paper improves high-fidelity dynamic scene reconstruction from blurry monocular videos through kinematics-aware 3D Gaussian Splatting. There are many potential societal consequences of our work, none of which we feel must be specifically highlighted here.

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

# Appendix Overview

# A. Method Details

### A.1. Detailed Procedure for Kinematic Basis Construction

The construction of the local kinematic orthonormal basis $\widetilde{\mathbf{R}} = [\, \mathbf{u}_x \mid \mathbf{u}_y \mid \mathbf{u}_z \,]$ is critical for disentangling motion-induced anisotropy from intrinsic shape deformation. This section provides the numerically stable implementation details of the kinematic basis construction, with particular attention to degenerate cases where the motion direction becomes ill-defined or colinear with a canonical reference axis.

Given the predicted instantaneous velocity vector $\mathbf{v}$, the primary motion axis $\mathbf{u}_z$ is obtained by normalizing the velocity direction. To ensure numerical stability in near-static regions, a small constant $\epsilon = 10^{-8}$ is added to the denominator:

$$\mathbf{u}_z = \frac{\mathbf{v}}{\|\mathbf{v}\| + \epsilon}. \tag{24}$$

To construct a stable orthonormal basis, we introduce a canonical reference direction $\mathbf{r}_{\text{ref}}$. Using a single fixed reference vector may lead to degeneracy when $\mathbf{u}_z$ becomes colinear with it. To avoid this issue, we deterministically select $\mathbf{r}_{\text{ref}}$ to maximize orthogonality with the motion direction:

$$\mathbf{r}_{\text{ref}} = \begin{cases} [0, 1, 0]^\top, & \text{if } |\mathbf{u}_z \cdot [1, 0, 0]^\top| > 0.99, \\ [1, 0, 0]^\top, & \text{otherwise.} \end{cases} \tag{25}$$

This selection guarantees that $\mathbf{r}_{\text{ref}}$ is never colinear with $\mathbf{u}_z$. Subsequently, the side axis $\mathbf{u}_x$ is constructed via the cross product between the primary motion axis and the reference direction:

$$\mathbf{u}_x = \frac{\mathbf{u}_z \times \mathbf{r}_{\text{ref}}}{\|\mathbf{u}_z \times \mathbf{r}_{\text{ref}}\| + \epsilon}. \tag{26}$$

This formulation directly yields a vector orthogonal to the motion direction and avoids the numerical instability associated with subtractive orthogonalization.

The third axis $\mathbf{u}_y$ is then defined to complete a right-handed coordinate system:

$$\mathbf{u}_y = \mathbf{u}_z \times \mathbf{u}_x. \tag{27}$$

The resulting kinematic rotation matrix is obtained by concatenating the three orthonormal axes:

$$\widetilde{\mathbf{R}} = [\, \mathbf{u}_x \mid \mathbf{u}_y \mid \mathbf{u}_z \,] \in SO(3). \tag{28}$$

This motion-aligned coordinate frame varies smoothly with the predicted velocity and provides a physically grounded reference for modeling motion-induced anisotropic deformation of dynamic Gaussian primitives.

## A.2. Gaussian Noise Injection for Deformation Stabilization

Monocular dynamic scene reconstruction is inherently ill-posed due to the ambiguity between object depth and motion, and the absence of multi-view geometric constraints. When trained directly with monocular supervision, deformation networks are prone to overfitting spurious high-frequency motion patterns at early optimization stages, often resulting in temporally unstable deformations and poor convergence.

To alleviate this issue, we inject Gaussian noise into the deformation network input during training. This stochastic perturbation serves as an implicit regularizer, encouraging the network to first capture smooth, low-frequency deformation trends before progressively refining fine-grained motion details as optimization proceeds.

### A.2.1. NOISE FORMULATION

As described in Sec. 4.1, for each canonical Gaussian primitive $\mathcal{G}$ with parameters $(\mathbf{x}, \mathbf{r}, \mathbf{s})$, the deformation network $\mathcal{F}_\theta$ predicts per-Gaussian offsets at time $t$ as:

$$(\delta\mathbf{x}_t, \delta\mathbf{r}_t, \delta\mathbf{s}_t) = \mathcal{F}_\theta(\mathbf{x}, \mathbf{r}, \mathbf{s}, \gamma(t) + \epsilon), \tag{29}$$

where $\gamma(t)$ denotes the temporal positional encoding, and $\epsilon \sim \mathcal{N}(0, \sigma_\epsilon^2(t))$ is a zero-mean Gaussian noise vector. The noise variance $\sigma_\epsilon^2(t)$ follows a deterministic decay schedule over training iterations, ensuring strong regularization at early stages and gradual annealing toward zero.

### A.2.2. LOG-LINEAR NOISE DECAY SCHEDULE

In practice, we adopt a monotonically decaying noise schedule inspired by learning-rate annealing strategies used in volumetric rendering methods. Specifically, the noise standard deviation at training iteration $k$ is defined as:

$$\sigma_\epsilon(k) = w(k) \cdot \left[\sigma_{\text{init}}(1 - \kappa) + \sigma_{\text{final}}\kappa\right], \quad \kappa = \text{clip}\left(\frac{k}{K_{\text{max}}}, 0, 1\right), \tag{30}$$

where $\sigma_{\text{init}}$ and $\sigma_{\text{final}}$ denote the initial and final noise magnitudes, respectively, and $K_{\text{max}}$ denotes the total number of iterations used for noise annealing.

To further stabilize early optimization, we optionally introduce a warm-up phase:

$$w(k) = \begin{cases} w_{\text{delay}} + (1 - w_{\text{delay}}) \sin\left(\frac{\pi}{2} \cdot \frac{k}{K_{\text{delay}}}\right), & k < K_{\text{delay}}, \\ 1, & \text{otherwise}, \end{cases} \tag{31}$$

where $w_{\text{delay}} \in (0, 1)$ controls the initial attenuation of noise and $K_{\text{delay}}$ denotes the duration of the warm-up phase.

The injected noise is applied only during training and serves to regularize deformation learning. Dynamic–static decomposition is subsequently performed based on the temporal variance of the predicted deformation offsets, and is therefore not directly affected by the injected noise. In practice, noise annealing converges well before deformation variance statistics are computed, ensuring a stable separation between dynamic and static Gaussian primitives. Empirically, we observe that removing noise annealing or maintaining a constant noise level degrades reconstruction quality, either by introducing persistent temporal jitter or by over-smoothing dynamic motion patterns. The decaying noise strategy thus provides an effective coarse-to-fine regularization mechanism for stabilizing monocular deformation learning.

## B. Training Details

Our framework is trained end-to-end for 30,000 iterations with a batch size of 2. We employ exponential decay schedulers for all learning rates. The position learning rate anneals from $1.6 \times 10^{-4}$ to $1.6 \times 10^{-6}$, while deformation, grid, and feature parameters follow similar schedules. Gaussian densification and pruning are performed using a gradient threshold of $1 \times 10^{-4}$ and an opacity threshold of $\alpha_{\text{min}} = 0.005$. Densification is applied periodically between iterations 500 and 10k. Dynamic scenes are modeled using a K-Planes representation with a spatial resolution of $64^3$ and a temporal resolution of 25. Temporal deformation is parameterized by a lightweight MLP with a width of 64 and a depth of 1. Additional regularization terms and weights follow standard practice. To stabilize optimization, we apply a global opacity reset at iteration 3k.

## C. Custom Dataset Details

We introduce DEOs, a custom dataset designed to evaluate monocular dynamic scene reconstruction involving non-rigid and elastic objects under realistic motion blur. To capture diverse physical behaviors, DEOs include two object categories: deformable objects (*e.g.*, soft foam dice and toy dolls) that undergo shape compression upon contact, and elastic objects (*e.g.*, basketballs, volleyballs, soccer balls, and rugby balls) that exhibit rapid shape recovery under dynamic interactions.

All sequences are captured in real-world indoor and outdoor environments, and feature motion blur induced by both object motion and camera movement. The dataset comprises eight sequences spanning rolling, bouncing, striking, and dribbling motions, as illustrated in Fig. 6, resulting in pronounced motion-induced ambiguity and non-rigid deformation.

These characteristics make DEOs particularly suitable for evaluating robustness to motion blur and physically meaningful non-rigid dynamics in monocular reconstruction. Compared to existing dynamic reconstruction benchmarks, which are limited to indoor scenes and rigid deformations, DEOs uniquely includes elastic objects captured under fast motion and realistic blur across diverse environments. A detailed comparison with BARD-GS (Lu et al., 2025) and NeRF-DS (Yan et al., 2023) is provided in Tab. 4.

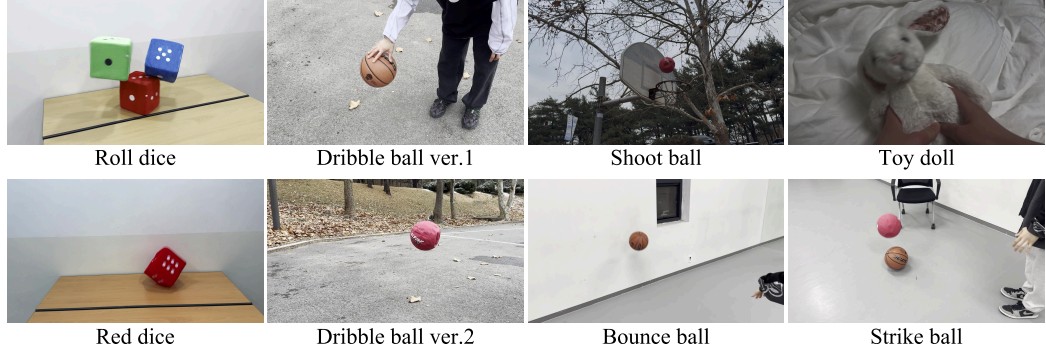

|  |  |  |  |
| :---: | :---: | :---: | :---: |
| Roll dice | Dribble ball ver.1 | Shoot ball | Toy doll |
| Red dice | Dribble ball ver.2 | Bounce ball | Strike ball |

*Figure 6.* Overview of the DEOs dataset, illustrating representative objects and scenes. The dataset includes both indoor and outdoor environments and features deformable and elastic objects undergoing fast motion with combined camera- and object-induced motion blur.

*Table 4.* Comparison of dynamic scene reconstruction benchmarks. DEOs features deformable and elastic objects undergoing fast motion with both camera-and object-induced motion blur, while existing benchmarks primarily focus on rigid or limited non-rigid motion and do not capture elastic deformation under realistic blur conditions.

| Dataset | Environment | Blur Source | | Motion Speed | Deformable Objects | Elastic Objects | # Cameras | # Scenes | FPS |
| --- | --- | --- | --- | --- | --- | --- | --- | --- | --- |
| | | Camera | Object | | | | | | |
| BARD-GS | Indoor | Yes | Yes | Fast | None | None | 2 | 12 | 24 / 120 |
| NeRF-DS | Indoor | Yes | No | Slow | Thin sheet | None | 2 | 8 | $\geq$30 |
| **DEOs** | Indoor + Outdoor | Yes | Yes | Fast | Colored foam dice, Toy doll | Soccer ball, Basketball, Volleyball | 2 | 8 | 60 |

# D. Full Results

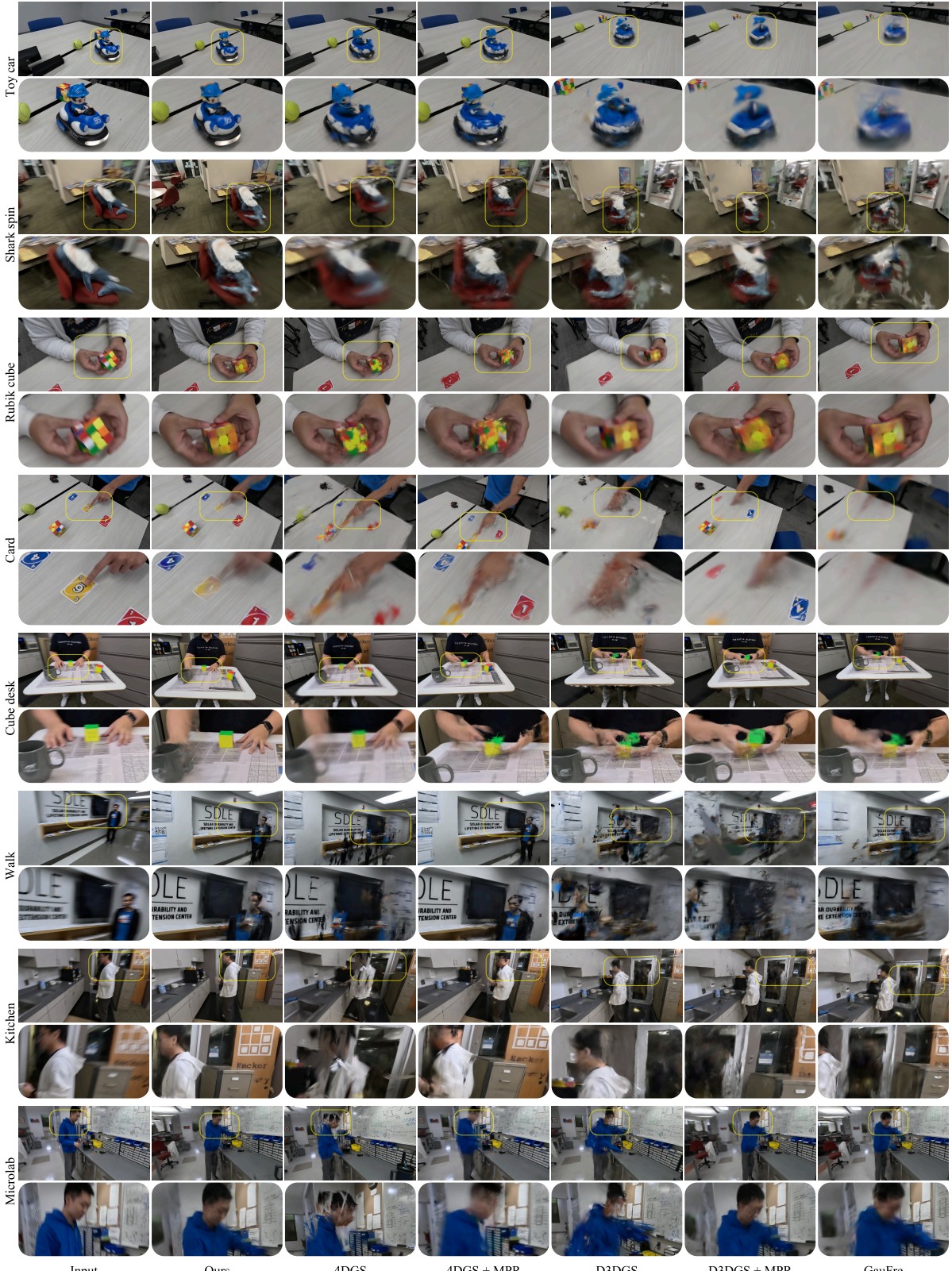

*Figure 7.* Per-scene qualitative comparison on the BARD-GS real-world blurry dataset.

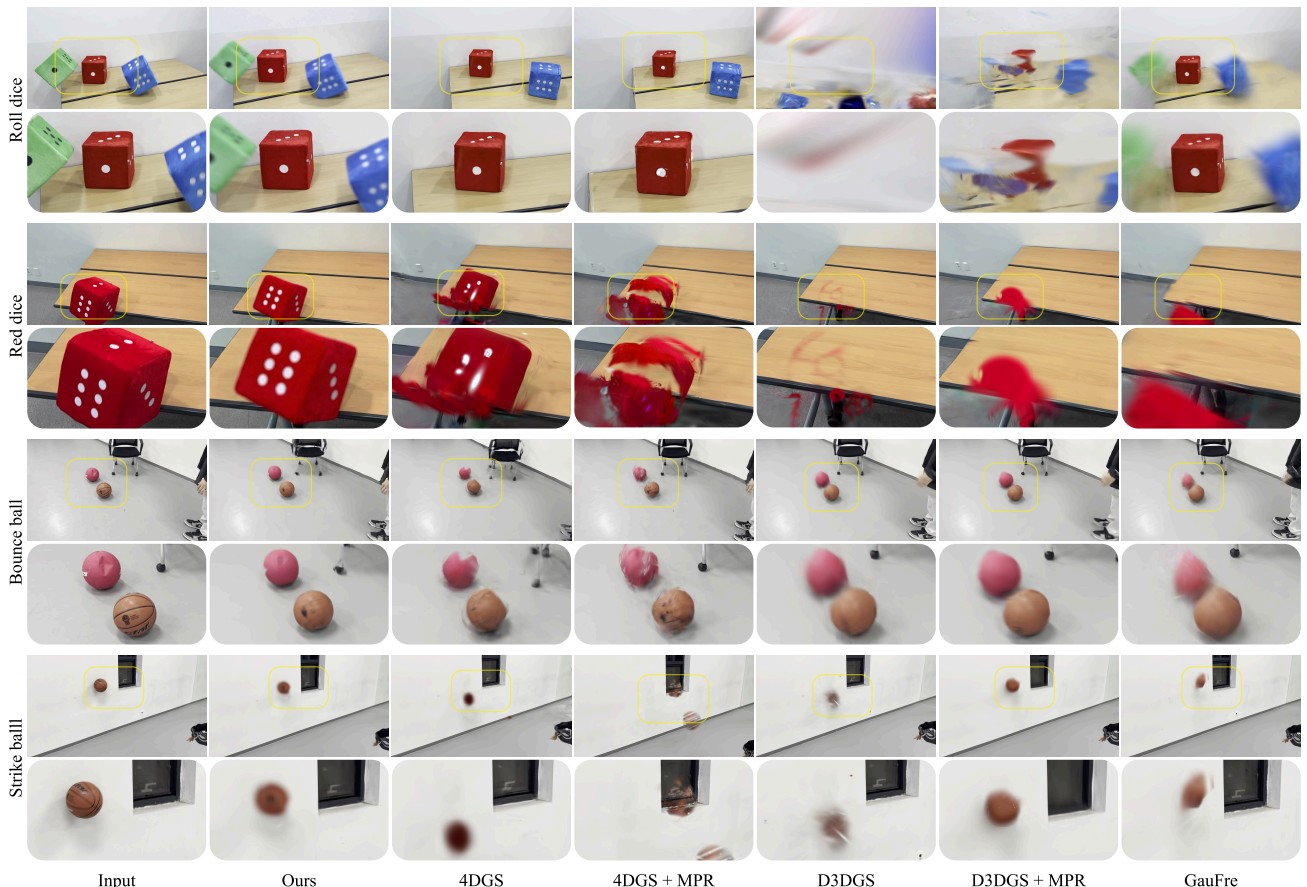

*Figure 8.* Per-scene qualitative comparison on the DEOs real-world blurry dataset.

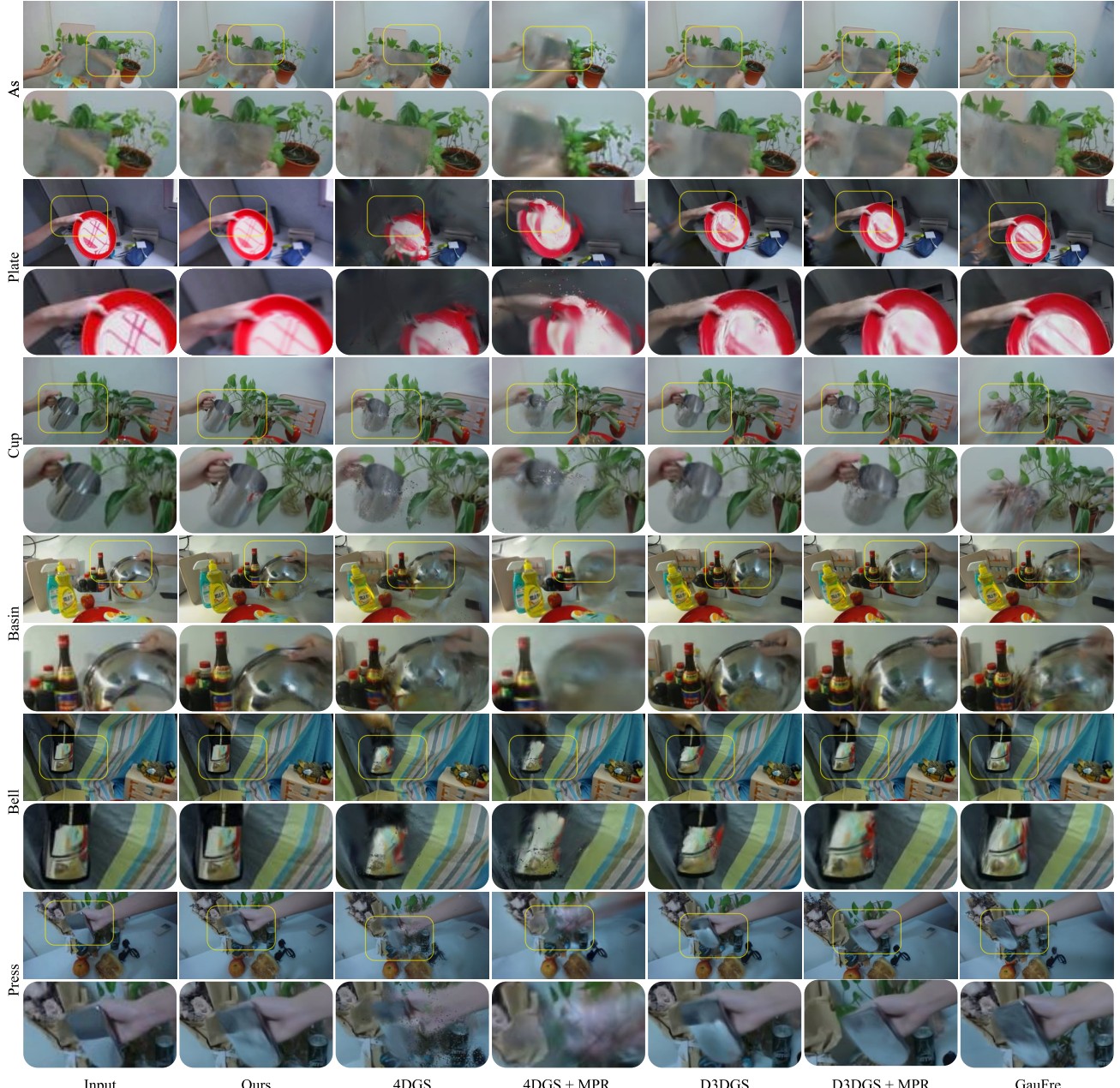

*Figure 9.* Per-scene qualitative comparison on the NeRF-DS dynamic specular iPhone dataset.

# E. Computational Efficiency

### E.1. Runtime Performance (FPS)

We evaluate the inference speed of our method compared to state-of-the-art dynamic scene reconstruction approaches. All measurements were conducted on a single NVIDIA RTX 3090 GPU. As shown in the Tables 5 to 7, our framework achieves significantly higher frame rates, consistently exceeding 300 FPS across most scenes.

This superior performance is attributed to our streamlined kinematic deformation formulation. Unlike methods that rely on heavy MLP evaluations for every query point or complex 4D feature interpolation, our approach analytically derives deformation from a compact set of motion parameters. This design minimizes the computational overhead per Gaussian, enabling high-fidelity dynamic rendering suitable for real-time applications.

*Table 5.* Quantitative comparison of rendering speed (FPS) on the BARD-GS dataset. Each color indicates the best , second best , and third best respectively.

| METHOD | BARD-GS | | | | | | | |
|---|---|---|---|---|---|---|---|---|
| | TOYCAR | RUBIKCUBE | CARD | CUBEDESK | SHARKSPIN | WALK | KITCHEN | MICROLAB |
| 4DGS+MPRNET | 20.982 | 18.433 | 24.451 | 24.421 | 29.695 | 26.315 | 22.902 | 31.001 |
| 4DGS | 14.596 | 22.100 | 27.615 | 18.820 | 29.611 | 25.480 | 29.713 | 25.237 |
| D3DGS+MPRNET | 53.308 | 68.130 | 35.750 | 55.236 | 30.409 | 28.042 | 48.770 | 33.197 |
| D3DGS | 71.306 | 53.507 | 56.991 | 55.573 | 38.078 | 23.401 | 47.556 | 27.895 |
| GAUFRE | 20.986 | 15.311 | 13.457 | 24.631 | 13.025 | 17.511 | 16.324 | 25.754 |
| **OURS** | 480.610 | 386.589 | 607.885 | 455.564 | 464.440 | 698.034 | 455.209 | 377.552 |

*Table 6.* Quantitative comparison of rendering speed (FPS) on the DEOs dataset.

| METHOD | DEOS | | | |
|---|---|---|---|---|
| | ROLLDICE | REDDICE | BOUNCE | STRIKE |
| 4DGS+MPRNET | 29.820 | 29.654 | 59.810 | 60.542 |
| 4DGS | 25.337 | 28.593 | 55.547 | 57.272 |
| D3DGS+MPRNET | 23.621 | 28.829 | 22.356 | 87.522 |
| D3DGS | 40.234 | 21.432 | 7.062 | 48.271 |
| GAUFRE | 18.026 | 19.857 | 17.490 | 12.523 |
| **OURS** | 403.363 | 380.001 | 250.392 | 420.1933 |

*Table 7.* Quantitative comparison of rendering speed (FPS) on the NeRF-DS dataset.

| METHOD | NERF-DS | | | | | |
|---|---|---|---|---|---|---|
| | AS | BASIN | BELL | CUP | PLATE | PRESS |
| 4DGS+MPRNET | 185.105 | 87.196 | 78.713 | 81.091 | 49.632 | 92.951 |
| 4DGS | 100.085 | 61.418 | 77.127 | 89.678 | 51.998 | 89.597 |
| D3DGS+MPRNET | 108.046 | 80.410 | 65.4847 | 96.097 | 92.817 | 106.878 |
| D3DGS | 115.153 | 104.217 | 79.790 | 99.943 | 123.137 | 125.690 |
| GAUFRE | 3.594 | 6.172 | 10.994 | 4.142 | 6.094 | 5.369 |
| **OURS** | 302.281 | 278.451 | 265.302 | 360.492 | 220.304 | 489.493 |

## E.2. Spatial Complexity (Number of Gaussians)

We evaluate the spatial complexity of our method compared to the baseline models in terms of the number of Gaussians in Tables 8 to 10. Our approach achieves a highly compact representation, utilizing significantly fewer Gaussian primitives than other methods. This efficiency stems from our model's ability to effectively identify and prune geometric redundancies often caused by motion-induced blur ambiguities. Consequently, we reduce the memory footprint without compromising reconstruction quality, demonstrating that our method successfully eliminates erroneous Gaussians while preserving fine-grained geometric and photometric details.

*Table 8.* Comparison of spatial complexity in terms of the number of Gaussians on the BARD-GS dataset. Our method achieves a compact representation by effectively eliminating redundant primitives, requiring significantly fewer Gaussians than baseline methods. Each color indicates the best , second best , and third best respectively.

| | BARD-GS | | | | | | | |
|---|---|---|---|---|---|---|---|---|
| **METHOD** | TOYCAR | RUBIKCUBE | CARD | CUBEDESK | SHARKSPIN | WALK | KITCHEN | MICROLAB |
| 4DGS+MPRNET | 54825 | 52762 | 69224 | 101226 | 50744 | 88359 | 155045 | 84793 |
| 4DGS | 82066 | 54664 | 71420 | 106047 | 48476 | 90150 | 175556 | 78291 |
| D3DGS+MPRNET | 70026 | 107022 | 142189 | 165279 | 266180 | 279899 | 196907 | 257004 |
| D3DGS | 79298 | 133790 | 113323 | 164138 | 203967 | 357026 | 194962 | 283734 |
| GAUFRE | 100456 | 121803 | 24666 | 178100 | 74790 | 113949 | 132610 | 291739 |
| **OURS** | 36651 | 43530 | 46929 | 33722 | 34829 | 60554 | 52390 | 62355 |

*Table 9.* Comparison of spatial complexity in terms of the number of Gaussians on the DEOs dataset.

| | DEOs | | | |
|---|---|---|---|---|
| **METHOD** | ROLLDICE | REDDICE | BOUNCE | STRIKE |
| 4DGS+MPRNET | 44730 | 43602 | 37552 | 85827 |
| 4DGS | 28080 | 50607 | 105948 | 95227 |
| D3DGS+MPRNET | 172147 | 115764 | 188924 | 75186 |
| D3DGS | 201634 | 262875 | 94534 | 90653 |
| GAUFRE | 82018 | 71338 | 81233 | 98844 |
| **OURS** | 16966 | 29244 | 15200 | 64527 |

*Table 10.* Comparison of spatial complexity in terms of the number of Gaussians on the NeRF-DS dataset.

| | NERF-DS | | | | | |
|---|---|---|---|---|---|---|
| **METHOD** | AS | BASIN | BELL | CUP | PLATE | PRESS |
| 4DGS+MPRNET | 55380 | 183399 | 148883 | 85764 | 192998 | 72883 |
| 4DGS | 54240 | 146115 | 103228 | 82097 | 211362 | 78601 |
| D3DGS+MPRNET | 125782 | 156263 | 158913 | 132703 | 128373 | 124999 |
| D3DGS | 111028 | 146462 | 194889 | 127092 | 120944 | 120855 |
| GAUFRE | 54881 | 97896 | 185277 | 66600 | 94454 | 84478 |
| **OURS** | 22210 | 26211 | 33034 | 42439 | 31360 | 31144 |

# F. Additional Ablation Results

*Table 11.* Additional per-scene ablation study on the BARD-GS dataset. Each color indicates the best , second best , and third best respectively.

| METHOD | CARD | | | SHARKSPIN | | | KITCHEN | | | MICROLAB | | |
|---|---|---|---|---|---|---|---|---|---|---|---|---|
| | PSNR↑ | SSIM↑ | LPIPS↓ | PSNR↑ | SSIM↑ | LPIPS↓ | PSNR↑ | SSIM↑ | LPIPS↓ | PSNR↑ | SSIM↑ | LPIPS↓ |
| OURS | 24.455 | 0.902 | 0.162 | 25.394 | 0.898 | 0.186 | 23.545 | 0.887 | 0.219 | 25.374 | 0.872 | 0.123 |
| W/O C.F. | 23.810 | 0.881 | 0.195 | 24.520 | 0.875 | 0.220 | 22.900 | 0.860 | 0.255 | 24.850 | 0.860 | 0.145 |
| W/O K.R. | 18.242 | 0.610 | 0.485 | 17.550 | 0.580 | 0.510 | 16.100 | 0.550 | 0.540 | 17.200 | 0.590 | 0.465 |
| W/O $\mathcal{L}_{\text{REG}}$ | 23.950 | 0.885 | 0.188 | 24.700 | 0.880 | 0.210 | 23.150 | 0.875 | 0.235 | 24.110 | 0.855 | 0.160 |
| W/O $\mathcal{L}_{\text{ANI}}$ | 24.100 | 0.890 | 0.180 | 24.820 | 0.882 | 0.205 | 23.010 | 0.870 | 0.245 | 25.020 | 0.865 | 0.138 |
| $\tau = 4e{-}5$ | 23.120 | 0.865 | 0.235 | 23.850 | 0.850 | 0.260 | 21.900 | 0.830 | 0.295 | 23.450 | 0.820 | 0.210 |
| $\tau = 1e{-}4$ | 17.850 | 0.710 | 0.420 | 18.200 | 0.705 | 0.445 | 16.500 | 0.680 | 0.470 | 18.100 | 0.690 | 0.435 |

*Table 12.* Additional per-scene ablation study on the DEOs dataset.

| METHOD | REDDICE | | | BOUNCE | | | STRIKE | | |
|---|---|---|---|---|---|---|---|---|---|
| | PSNR↑ | SSIM↑ | LPIPS↓ | PSNR↑ | SSIM↑ | LPIPS↓ | PSNR↑ | SSIM↑ | LPIPS↓ |
| OURS | 23.405 | 0.836 | 0.167 | 25.256 | 0.956 | 0.051 | 22.345 | 0.920 | 0.117 |
| W/O C.F. | 22.140 | 0.812 | 0.198 | 24.810 | 0.948 | 0.059 | 20.880 | 0.895 | 0.145 |
| W/O K.R. | 16.850 | 0.620 | 0.440 | 18.110 | 0.720 | 0.365 | 15.920 | 0.680 | 0.395 |
| W/O $\mathcal{L}_{\text{REG}}$ | 22.950 | 0.825 | 0.175 | 23.950 | 0.920 | 0.084 | 21.940 | 0.911 | 0.124 |
| W/O $\mathcal{L}_{\text{ANI}}$ | 23.110 | 0.830 | 0.170 | 25.040 | 0.952 | 0.055 | 22.150 | 0.916 | 0.120 |
| $\tau = 4e{-}5$ | 20.850 | 0.790 | 0.230 | 22.420 | 0.890 | 0.115 | 19.640 | 0.840 | 0.198 |
| $\tau = 1e{-}4$ | 17.110 | 0.695 | 0.405 | 18.350 | 0.790 | 0.330 | 16.220 | 0.755 | 0.375 |

*Table 13.* Additional per-scene ablation study on the NeRF-DS dataset.

| METHOD | BASIN | | | CUP | | | PLATE | | | PRESS | | |
|---|---|---|---|---|---|---|---|---|---|---|---|---|
| | PSNR↑ | SSIM↑ | LPIPS↓ | PSNR↑ | SSIM↑ | LPIPS↓ | PSNR↑ | SSIM↑ | LPIPS↓ | PSNR↑ | SSIM↑ | LPIPS↓ |
| OURS | 22.395 | 0.802 | 0.159 | 24.930 | 0.902 | 0.100 | 20.938 | 0.819 | 0.218 | 25.909 | 0.870 | 0.190 |
| W/O C.F. | 21.850 | 0.785 | 0.185 | 24.750 | 0.898 | 0.104 | 20.140 | 0.795 | 0.250 | 24.850 | 0.850 | 0.220 |
| W/O K.R. | 16.540 | 0.610 | 0.420 | 18.240 | 0.720 | 0.385 | 15.850 | 0.640 | 0.495 | 19.120 | 0.680 | 0.440 |
| W/O $\mathcal{L}_{\text{REG}}$ | 22.010 | 0.791 | 0.168 | 24.620 | 0.894 | 0.108 | 20.550 | 0.810 | 0.228 | 25.400 | 0.861 | 0.199 |
| W/O $\mathcal{L}_{\text{ANI}}$ | 22.140 | 0.796 | 0.165 | 24.110 | 0.881 | 0.125 | 20.720 | 0.814 | 0.224 | 25.610 | 0.866 | 0.195 |
| $\tau = 4e{-}5$ | 20.820 | 0.760 | 0.225 | 23.150 | 0.855 | 0.180 | 19.220 | 0.775 | 0.315 | 23.450 | 0.820 | 0.270 |
| $\tau = 1e{-}4$ | 17.110 | 0.672 | 0.425 | 18.410 | 0.770 | 0.385 | 16.200 | 0.690 | 0.495 | 18.750 | 0.730 | 0.470 |

