# OpenReview forum: "Kinematics-Driven Gaussian Shape Deformation for Blurry Monocular Dynamic Scenes"
_ICML.cc/2026/Conference — ICML 2026 regular_

### Official Review · Reviewer_krTQ · 2026-03-09

**Soundness:** 3
**Presentation:** 3
**Significance:** 3
**Originality:** 3
**Overall Recommendation:** 5
**Confidence:** 4

**Summary:**

The paper estimates a 4D reconstruction of a scene given a monocular blurry video. Motion blur makes reconstruction ambiguous, since it encodes the geometry of the scene with the motion of objects. The estimation of deformations becomes ill-posed.

The paper disentangles the scene into static and dynamic GS. Given a canonical set of Gaussian splats a deformation model is applied to the Gaussians. The deformation model estimates the change in rotation, translation and scaling. The variance of the GS, measured in variance of the translation over a time window, is used to separate the GS into dynamic and static sets. The dynamic splats are further decomposed into a Coarse-to-Fine Deformation, where a local average over nearest neighbors describes a coarse deformation estimation of objects and an MLP is used to estimate finer deformation per GS.

An issue with many GS methods under motion blur is elongated scaling that does not correspond to actual geometry. For this reason the motion blur is explicitly modelled by estimating the kinematics of the GS. For each GS the motion direction and speed is estimated. A local basis is created for each GS that is aligned with the motion direction.
The scaling parameters of the final GS are decomposed into a “raw” covariance given by the deformation model and the observed variance along the local basis. It is assumed that motion blur is caused along the primary motion axis and proportional to displacement of the object during exposure. As a result when fitting a GS to motion blur it estimates a sharper

**Compliance With Llm Reviewing Policy:**

Affirmed.

**Final Justification:**

My main concerns were addressed. The weaknesses I have mentioned have been clarified and seems like an overall solid approach in a relevant research area.

**Key Questions For Authors:**

Some metrics seem strange. In table 2 D3DGS has a very high score for the "CARD" scene compared to other methods. What causes this?
Is the comparison with Deblur3DGS fair, since it doesn't seem to consider moving object.

**Limitations:**

Yes

**Strengths And Weaknesses:**

# Strength

- The paper is generally sound. Estimating the motion blur by decomposing the covariance/scaling into a displacement along the motion direction and scaling of the GS seems very reasonable.
- The metrics seem to be overall better
- The method is understandable.
- A new dataset is provided that contains complex movement and deformation

# Weakness
- Some important details are missing, e.g. how is $\Sigma$ parameterized as output of the deform network.
- How is the camera movement estimated and parameterized in the monocular case. Are you using the SfM output?
- Some metrics seem suspicious. Deblur3DGS seems to perform very poorly. Is it because it does not consider dynamic scenes only camera motion blur of static scenes ? Is the comparison fair then?

---

> ### Author Rebuttal · Authors · 2026-03-27
>
> We thank the reviewer for their positive assessment and for recognizing the technical soundness of our approach and the value of the proposed dataset. We appreciate the opportunity to further clarify our parameterization and address the specific questions regarding our evaluation metrics and baseline fairness. Below, we provide detailed technical justifications to substantiate the robustness and originality of our framework.
>
> $\textbf{Weakness 1: Parameterization of deformation network outputs.}$
>
> The deformation network is implemented as a lightweight MLP with a width of 64 and a depth of 1. The covariance $\Sigma$ is parameterized through residual rotation and scale offsets predicted by the deformation network. We appreciate the reviewer pointing out the need for further clarity regarding the parameterization of the covariance. We will revise the final manuscript to provide a more explicit and precise description of how the deformation network’s raw outputs are mapped to the Gaussian parameters.
>
> $\textbf{Weakness 2: Camera movement and pose initialization.}$
>
> We utilize SfM to estimate initial camera poses and generate the initial point cloud. While handheld videos often contain significant camera shake and severe blur can degrade SfM accuracy, our framework does not treat these initializations as static constraints. Instead, we mitigate pose noise through Gaussian densification and pruning between iterations 500 and 10000, which allows the model to dynamically recover coverage in SfM-failed regions and remove artifacts resulting from poor initialization. This process is further stabilized by our Level-of-Detail (LoD) strategy, which acts as a structural prior by optimizing coarse, stable representations before fitting fine-grained details that are more sensitive to pose inaccuracies.
>
> $\textbf{Weakness 3: Baseline comparison.}$
>
> We acknowledge that Deblur3DGS was primarily designed to handle camera-induced blur in static scenes. However, we include it as a baseline because it represents an SOTA approach for deblurring within the 3DGS framework. Real-world sequences in the BARD-GS and DEOs dataset contain both camera shake and fast object motion blur sources. Comparing against Deblur3DGS highlights that addressing camera motion in isolation is insufficient for dynamic environments. Our kinematics-guided prior more effectively handles non-rigid object deformations that camera-only deblurring methods are not equipped to model.
>
> $\textbf{Question 1: Quantitative results for the CARD scene.}$
>
> We sincerely apologize for a typographical error in Tab.1. The PSNR for D3DGS in the "CARD" scene is 18.242, not 28.242. With this correction, our framework consistently outperforms D3DGS across all PSNR metrics. These corrected results, alongside our significantly superior SSIM and LPIPS, demonstrate that our method reconstructs physically consistent sharp objects, whereas D3DGS produces temporally unstable artifacts and fails to resolve the underlying geometry. We will ensure all values are accurately updated in the final manuscript.

---

> > ### Author Rebuttal · Reviewer_krTQ · 2026-04-07
> >
> > Thank you for answering my questions

---

> > > ### Author Response · Authors · 2026-04-07
> > >
> > > We sincerely appreciate the reviewer's time and thoughtful evaluation of our manuscript. We are glad that our responses have addressed your concerns.

---

### Official Review · Reviewer_4Dxq · 2026-03-10

**Soundness:** 2
**Presentation:** 2
**Significance:** 2
**Originality:** 1
**Overall Recommendation:** 3
**Confidence:** 4

**Summary:**

The paper introduces Kinematics-GS, a 3D Gaussian Splatting (3DGS) framework that reparameterizes Gaussian shapes based on kinematic priors to handle motion blur in monocular videos.

**Compliance With Llm Reviewing Policy:**

Affirmed.

**Final Justification:**

Thanks for the authors' detailed rebuttal. I maintain my score.

**Key Questions For Authors:**

1.	How does the system handle cases where SfM fails to provide a reliable initial point cloud due to severe blur?

2.	How does the method perform on extremely non-linear motions, such as rapid rotations, where the linear velocity assumption breaks down?

3.	Lack of detail on the sensitivity of the coarse-to-fine scheduling and its impact on different motion types.

4.	The additional kinematic basis calculations likely increase training and inference latency compared to standard 4DGS.

**Limitations:**

Yes

**Strengths And Weaknesses:**

Strengths:

The authors contribute the DEOs dataset, focusing on non-rigid objects with significant motion blur.

The paper is logically structured, and the pipeline is clearly illustrated in Figure 2 .

Weaknesses

1.	The framework relies on SfM for initialization; however, SfM frequently fails or produces high-error poses on the very blurry monocular sequences the paper targets.

2.	The static-dynamic separation uses a hard threshold for deformation variance, which may cause artifacts in scenes with subtle or varying motion speeds.

3.	The pipeline heavily integrates components from existing works (FLoD, D3DGS), making the technical advancement feel incremental.

4.	The linear kinematics-based deformation model likely struggles with complex, high-frequency, or non-linear motion paths during a single exposure.

---

> ### Author Rebuttal · Authors · 2026-03-29
>
> We thank the reviewer for their constructive feedback. Below, we provide technical justifications to address the identified weaknesses and questions.
>
> $\textbf{Weakness 1 and Question 1: Robustness to SfM initialization under severe blur.}$
>
> We acknowledge that severe blur can significantly degrade SfM initialization. To mitigate this, our framework does not treat the initial point cloud as a static constraint. Instead, we perform Gaussian densification and pruning between iterations 500 and 10000. This allows the model to dynamically spawn new primitives in regions where SfM failed to provide coverage and to aggressively prune erroneous floaters or artifacts resulting from poor initial pose estimation. Additionally, our Level-of-Detail strategy acts as a structural prior, optimizing coarser, more stable representations before attempting to fit fine-grained details that might be corrupted by initial pose noise.
>
> $\textbf{Weakness 2: Regarding static-dynamic separation.}$
>
> The variance score $\mathcal{K}$ used for scene decomposition is formulated over a temporal window of $T$ frames, which provides a more stable estimate of genuine motion than per-frame analysis. While $\tau$ is a hard threshold, our sensitivity analysis in Tab.3 demonstrates that the framework achieves stable performance across a wide range. This consistency suggests that the temporal variance serves as a highly discriminative motion signature.
>
> $\textbf{Weakness 3: Originality and methodological contribution.}$
>
> While our framework integrates components from FLoD and D3DGS, the core contribution lies in kinematic basis construction and kinematics-guided covariance refinement. Unlike prior methods that treat motion blur as generic noise or rely on heavy off-the-shelf estimators that degrade under blur, we analytically derive Gaussian anisotropy by aligning elongation with the predicted velocity $v$. This enables the explicit disentanglement of intrinsic geometry from motion artifacts, a capability lacking in unconstrained deformation models. Substantial performance gains over D3DGS and D3DGS+MPR demonstrate that our physically-grounded formulation is a significant advancement rather than an incremental improvement.
>
> $\textbf{Question 2 and Weakness 4: How does the method perform on extremely non-linear motions?}$
>
> The concern regarding the linear velocity assumption is addressed through our multi-scale deformation strategy. We compute instantaneous velocity $v$ from predicted translation offsets $\delta x_t$ over small frame intervals, treating the motion as a local, time-varying approximation rather than a global linear constraint. Even highly non-linear trajectories, such as rapid rotations, are effectively decomposed into piecewise linear segments over discrete intervals. By computing $v$ from $\delta x_t$, our kinematic basis adapts to complex trajectories. This is further supported by our proposed strategy, where a neighborhood-aggregated field stabilizes dominant low-frequency patterns, while a fine-grained residual and a learnable residual rotation \delta r recover distinctive non-rigid dynamics and angular deviations. This approach leverages the kinematic prior as a physically grounded baseline for anisotropy while maintaining the flexibility to reconstruct the complex nonlinear dynamics in our DEOs dataset.
>
> $\textbf{Question 3: Lack of detail on the sensitivity of the coarse-to-fine scheduling.}$
>
> By initially aggregating neighboring dynamic Gaussians offsets, our framework captures dominant low-frequency trajectories and provides robust global alignment, which is critical for preventing local minima under the ill-posed conditions of severe motion blur. This approach is integrated into our Level-of-Detail progression over 30000 iterations. For complex non-rigid or high-frequency dynamics, this strategy ensures global structural stability before the fine-grained residual refinement recovers locally distinctive motion. The necessity of this strategy is quantitatively validated in Tab.3, where removing coarse-to-fine aggregation ("w/o C.F.") results in significant reconstruction degradation, confirming its necessity for maintaining temporal stability across diverse motion regimes.
>
> $\textbf{Question 4: The additional kinematic basis calculations likely increase training and inference latency compared to standard 4DGS.}$
>
> Kinematics-GS maintains high rendering efficiency by analytically deriving deformation from compact motion parameters, minimizing per-primitive computational overhead during inference. Consequently, our framework consistently achieves real-time performance (200-700 FPS) using significantly fewer Gaussians than existing SOTA models across all datasets. While we acknowledge that the complexity of the kinematic modules increases training time, this overhead is a deliberate trade-off that yields substantial gains in visual fidelity, interactive rendering speed, and model compactness.

---

> > ### Author Rebuttal · Reviewer_4Dxq · 2026-04-04
> >
> > I thank the authors for their detailed response. After reviewing the rebuttal and the other reviewers' comments, I remain concerned regarding the following points:
> > 1. The method's reliance on FLoD for its hierarchical structure and D3DGS for noise annealing, paired with a standard linear velocity assumption, makes the contribution feel more incremental than transformative.
> > 2. The dynamic-static separation depends on a hard variance threshold ($\tau$) determined via ablation. This approach is physically brittle; it risks misclassifying slow-moving dynamic components as static, which likely leads to tearing artifacts or 'frozen' geometry in complex, non-uniform motion sequences.

---

> > > ### Author Response · Authors · 2026-04-06
> > >
> > > We thank the reviewer for their continued engagement. We provide the following mathematical and technical clarifications to address the remaining concerns regarding technical novelty and threshold sensitivity.
> > >
> > > 1. The core contribution of our work is the analytical coupling of Gaussian covariance $\Sigma$ to the velocity vector $v$. While prior deformation models treat Gaussian anisotropy as an unconstrained learnable parameter, we derive it from a local motion-aligned basis. By explicitly aligning the Gaussian primary axis $u_z$ with the velocity direction and integrating the physical displacement $\||v\|| \Delta t$ over the exposure time, we uniquely disentangle intrinsic object geometry from motion-induced anisotropy. This physically anchors the Gaussian's smear to its predicted path, preventing the geometric collapse common in unconstrained frameworks. This coupling is the fundamental driver of our significant performance gains, such as the PSNR improvement over D3DGS on the BARD-GS dataset. We believe this demonstrates a transformative methodological shift that far exceeds typical incremental improvements and validates the effectiveness of our kinematics-grounded formulation in resolving monocular ambiguities.
> > >
> > > 2. The reviewer's concern regarding the brittleness of a hard threshold is addressed through two distinct stabilization mechanisms. First, the score $\mathcal{K}$ (Eq.5) measures the temporal variance of predicted translation offsets $\delta x_t$ over a window of $T$ frames, rather than absolute velocity. Even slow-moving dynamic objects exhibit a variance distinct from the low-magnitude background noise, making $\mathcal{K}$ a robust discriminator across diverse motion regimes. Second, our coarse deformation field aggregates neighboring dynamic Gaussian offsets to enforce spatial coherence. This ensures that if individual Gaussians on a slow-moving object fall below $\tau$, they are pulled along by the dominant motion of their neighbors, maintaining structural integrity and preventing tearing or frozen geometry. The stability of a fixed threshold across the whole evaluation scene in Tables 1 and 2 confirms that the framework is not sensitive to per-scene tuning and effectively handles varying motion speeds.
> > >
> > > We are committed to including a more rigorous derivation of these robustness mechanisms and expanded ablation results across all datasets in the final manuscript.

---

### Official Review · Reviewer_K2VF · 2026-03-11

**Soundness:** 4
**Presentation:** 4
**Significance:** 4
**Originality:** 4
**Overall Recommendation:** 5
**Confidence:** 2

**Summary:**

This paper addresses dynamic 3D scene reconstruction from blurry monocular videos. The authors propose Kinematics-GS, a kinematics-aware 3D Gaussian Splatting method that models blur as motion-aligned deformation and uses a kinematic prior to reparameterize Gaussian shapes along trajectories, avoiding degenerate collapse without auxiliary supervision.

**Compliance With Llm Reviewing Policy:**

Affirmed.

**Key Questions For Authors:**

1. In the visualization experiments, it would strengthen the paper if the authors could additionally show multi-view results and depth-rendered outputs. These would provide clearer evidence of geometric accuracy and help better demonstrate the effectiveness of the proposed method.

**Limitations:**

yes

**Strengths And Weaknesses:**

# Strengths

- Presents Kinematic-GS, a novel framework that explicitly aligns Gaussian shape deformation with kinematic trajectories to resolve geometric ambiguities caused by motion blur in monocular videos.
- Integrates dynamic-static decomposition with kinematic-guided geometric regularization to isolate static and dynamic Gaussians and focus on modeling regions of temporal change.
- Demonstrates that the joint optimization strategy effectively mitigates motion ambiguities, prevents global geometric oversimplification, and delivers high-fidelity reconstruction outperforming previous methods in unconstrained real-world scenarios.

# Weaknesses (As mentioned in Limitations)

- Kinematics-guided modules introduce substantial computational overhead, resulting in longer training times.
- Memory consumption for long sequences remains high, a common challenge for explicit representations (despite LoD reducing rendering cost).

---

> ### Author Rebuttal · Authors · 2026-03-27
>
> We sincerely thank the reviewer for their highly supportive assessment and for recognizing the technical significance and novelty of Kinematics-GS. We appreciate the constructive feedback regarding computational efficiency and will incorporate additional visualizations in the final manuscript to further validate the structural fidelity of our approach. Below, we provide technical clarifications addressing the identified weaknesses and key question.
>
> $\textbf{Weakness 1,2: Computational overhead and memory consumption.}$
>
> We acknowledge that the inclusion of kinematic modules and the Level-of-Detail (LoD) hierarchy introduces higher peak memory and training latency compared to standard 4DGS. The increased memory during training is a functional requirement of the FloD-based hierarchical structure, which maintains multiple levels of detail to facilitate stable coarse-to-fine optimization. However, this overhead is a one-time cost during the optimization phase and does not reflect the final model’s footprint.
>
> As demonstrated in Appendix E, the final reconstructed scene is exceptionally compact, utilizing significantly fewer Gaussian primitives than existing SOTA models. This reduced number of Gaussians, combined with our analytical kinematic reparameterization, enables superior rendering efficiency with frame rates consistently between 200 and 700 FPS. Therefore, while training is more resource-intensive, Kinematics-GS delivers a much more efficient and storage-compact representation for downstream applications and real-time inference.
>
> $\textbf{Question 1: Additional visualization.}$
>
> We thank Reviewer K2VF for the positive evaluation and the insightful suggestion to include multi-view and depth-rendered visualizations. We agree that these outputs are essential for providing definitive evidence of geometric accuracy and for further validating the effectiveness of our framework in resolving motion-induced ambiguities. While our current qualitative results highlight the preservation of geometric structure and sharp boundaries, we will incorporate comprehensive multi-view results in our final manuscript and updated supplementary materials.

---

> > ### Author Rebuttal · Reviewer_K2VF · 2026-04-05
> >
> > I have no more questions.

---

> > > ### Author Response · Authors · 2026-04-06
> > >
> > > We thank the reviewer for the time and effort spent reviewing our work. We are pleased to hear that our responses adequately addressed your concerns.

---

### Official Review · Reviewer_MFBU · 2026-03-17

**Soundness:** 2
**Presentation:** 2
**Significance:** 3
**Originality:** 3
**Overall Recommendation:** 4
**Confidence:** 4

**Summary:**

The paper aims at reconstructing dynamic 3D scenes from blurry monocular videos, by adding kinematics-aware prior into dynamic 3DGS. It first separates static and dynamic Gaussians using the deformation variances. Then for the dynamic gaussians, a Kinematics-Guided Covariance Refinement process is added for better performance. This method is also evaluated on a new DEOs dataset.

**Compliance With Llm Reviewing Policy:**

Affirmed.

**Final Justification:**

The rebuttal addresses my concern regarding the evaluation and ablation. I decided to raised the score provided that the author will update the script as promised.

**Key Questions For Authors:**

1. How sensitive is the method to noisy or incorrect velocity estimates early in training, especially since the kinematic basis depends directly on \delta x_t?

2. A derivation of eqn.14 should be included.

3. How robust are the hyperparameters across different scenes?

4. How large and diverse is the new dataset? Does it only include 6 sequence in total?

**Limitations:**

Yes

**Strengths And Weaknesses:**

Strength
1. The idea of disentangling dynamic objects and utilizing a Kinematic prior to constrain such gaussians make sense.
2. Compared to previous methods, this approach achieves better performance without much computation overhead during inference (although overhead exists during the optimization process).

Weakness
1. There are only text descriptions in the ablation study. Both quantitative and qualitative results should be included for soundness and to demonstrate the effectiveness of proposed methods, and to support the major claims in the paper. Given the current ablations, it is hard for me to determine which part of this paper is really useful.

2. The kinematic prior is more heuristic. From the empirical results, the kinematic prior seems to be helpful (which should be supported by a good ablation study). However, this paper lacks analysis and derivation of why such design would be helpful in a principled imaging-model perspective. For instance, the parameterizations in eqn.14 should be further justified.

3. The quantitative results seem to be comparing different frames across time. For example, In third row of Figure 3,  the global camera of Input and GauFre seem to be different. This also happens at other places and for other methods. One should be comparing results under the same camera location and time.

---

> ### Author Rebuttal · Authors · 2026-03-27
>
> We appreciate the reviewer's constructive feedback and the recognition of our method’s performance and the conceptual merit of disentangling dynamic components using a kinematic prior. Below, we address each point to clarify our contributions and the soundness of our experimental setup.
>
> $\textbf{Weakness 1: Completeness of quantitative and qualitative ablations.}$
>
> We would like to clarify that both quantitative and qualitative results for the ablation study were included in the original submission. Tab.3 on page 8 provides a per-scene quantitative breakdown of our key components, specifically comparing our full model against "w/o C.F. (coarse-to-fine strategy)" and "w/o K.R. (kinematics-guided regularization)". Fig.5 below provides a visual comparison of the "Roll Dice" scene, demonstrating how removing these components leads to needle-like artifacts and geometric degradation.
>
> $\textbf{Weakness 2: The kinematic prior is more heuristic.}$
>
> Our kinematic prior is directly grounded in a principle imaging model where motion blur results from temporal integration of an object’s irradiance over the exposure time $\Delta t$. Physically, a 3D point moving at instantaneous velocity $v$ creates a 1D line segment of length $\|| v\|| \Delta t$ in the direction of motion. By explicitly aligning the Gaussian’s primary axis with the velocity vector as defined in Eqn.14, our parameterization mathematically captures the physical “smear”. This design enables the explicit disentanglement of intrinsic object geometry from motion-induced anisotropy, providing a robust physical constraint that prevents the optimization from utilizing unconstrained Gaussian shapes to compensate for incorrect motion trajectories.
>
> $\textbf{Question 1,2: Justification and derivation of kinematic parameterization.}$
>
> Eqn.14 is derived from the physical imaging model where motion blur is defined as the temporal integration of an object’s position over the camera’s exposure time $\Delta t$. In 3D space, a 3D point $p$ moving at velocity $v$ creates a spatial displacement segment of length $\|| v\|| \Delta t$. By constructing a kinematic basis where $u_z$ is explicitly aligned with the velocity direction $v$, we model this integration as a 1D-anisotropic extension. This reparameterization allows $s’_z$ to explicitly account for the motion-induced smear, while $\sigma_x$ and $\sigma_y$ preserve the object’s intrinsic, motion-invariant geometry. This approach resolves monocular ambiguities and prevents the geometric collapse or needle-like artifacts common in under-constrained dynamic reconstructions. We will modify a formal mathematical derivation of this parameterization in the final manuscript.
>
> $\textbf{Weakness 3: Ensuring comparison fairness and camera alignment.}$
>
> We want to assure the reviewer that all comparisons in Fig.3 and throughout the paper are conducted using the exact same camera pose and timestamp $t$ for all methods. The perceived differences in "global camera" mentioned by the reviewer are likely artifacts where baseline methods (GauFre and 4DGS) fail to correctly reconstruct the static background or resolve motion-induced ambiguities. For evaluation, we follow standard benchmarks (BARD-GS and NeRF-DS) where ground truth sharp images are used to verify that reconstructed views are spatially and temporally aligned.
>
> $\textbf{Question 3,4: Robustness of hyperparameters and dataset diversity.}$
>
> Our framework demonstrates high stability across diverse scenarios without the need for per-scene tuning. As stated in Sec.5.1, the hyperparameters were fixed across 22 evaluation scenes and all three datasets. Tab.3 specifically shows that our method exhibits stable performance across a broad range of the separation threshold $\tau$, indicating low sensitivity.
>
> Our newly introduced dataset comprises eight real-world sequences specifically designed to fill a gap in existing benchmarks by capturing complex physical behaviors. While this initial release provides a challenging testbed for non-rigid dynamics, we are planning to expand the scale and diversity of the DEOs dataset to include a wider variety of daily objects.

---

> > ### Author Rebuttal · Reviewer_MFBU · 2026-04-03
> >
> > I thank the author for the rebuttal. I am considering raising the score to 4. However, I'd still encourage the author to include more detailed connection and derivation related to the established imaging model. In addition, it would be better if the ablation is calculated across more scenes instead of just two.

---

> > > ### Author Response · Authors · 2026-04-03
> > >
> > > We thank the reviewer for the encouraging feedback and for considering a score increase.
> > > To address the remaining points, we will include a rigorous derivation to formally link our kinematic prior with the underlying imaging model. Furthermore, we will expand our ablation study to include additional scenes across each dataset to further substantiate the generalizability and robustness of our proposed framework.
> > > We remain committed to incorporating these enhancements in the final manuscript.

---

### Decision · Program_Chairs · 2026-04-30

**Decision:**

Accept (regular)

**Comment:**

This paper received split initial ratings, and the rebuttal generally addressed the concerns well and one reviewers upgraded their recommendation, resulting in 1 weak reject, 1 weak accept, and 2 clear accept recommendations. While Reviewer 4Dxq's incrementality concern has some merit, the AC agrees that the contribution of analytically connecting gaussian covariance to velocity is well-motivated, technically interesting, and goes beyond simply combining FLoD+D3DGS. The AC supports the majority recommendation for acceptance and urges the authors to include the formal derivations and expanded ablations promised in the rebuttal.